# What Makes ImageNet Look Unlike LAION

**Ali Shirali**                                                                                              *shirali__ali@berkeley.edu*
*University of California, Berkeley*

**Moritz Hardt**
*Max Planck Institute for Intelligent Systems, Tübingen and Tübingen AI Center*

**Reviewed on OpenReview:** *https://openreview.net/forum?id=IrBYuh9W3T*

## Abstract

ImageNet was famously created by querying several image search engines such as Flickr. What if we recreated ImageNet instead by searching the massive LAION dataset based on image captions alone? In this work, we carry out this counterfactual investigation. We find that the resulting ImageNet recreation, which we call LAIONet, looks distinctly unlike the original. Specifically, the intra-class similarity of images in the original ImageNet is dramatically higher than it is for LAIONet. Consequently, models trained on ImageNet perform significantly worse on LAIONet. We propose a rigorous explanation for the discrepancy in terms of a subtle, yet important, difference in two plausible causal data-generating processes for the respective datasets, that we support with systematic experimentation. In a nutshell, searching based on an image caption alone creates an information bottleneck that mitigates the selection bias otherwise present in image-based filtering. Our explanation formalizes a long-held intuition in the community that ImageNet images are stereotypical, unnatural, and overly simple representations of the class category. At the same time, it provides a simple and actionable takeaway for future dataset creation efforts.

## 1 Introduction

For nearly a decade, ImageNet (Deng et al., 2009) was the focal benchmark for much of computer vision and deep learning. Created from image web search results and human filtering, ImageNet contributed curated images suitable for supervised learning at the time. In recent years, however, the community has seen a new generation of models trained on massive amounts of noisy image-text data gathered from the web with minimal curation. Available to the academic public is the massive scale LAION dataset, in two versions, featuring 400 million (Schuhmann et al., 2021) and 5 billion (Schuhmann et al., 2022) crawled image-text pairs, filtered by the OpenAI CLIP model (Radford et al., 2021) for sufficient image-text relevance rather than human annotators.

At the outset, LAION works much like text-based web image search. We can specify a query and retrieve images with high similarity between the query and the text surrounding the image on the website from which it was crawled. We can therefore search LAION for each of the 1000 categories in the ImageNet ILSVRC-2012 dataset[1] and retrieve images corresponding to each of the classes. This process is much like the first step of creating ImageNet from Flickr search results, except that LAION replaces Flickr, but either way, both are based on web crawls. Where the creators of ImageNet hired human annotators to filter images, we analyze image captions to ensure that the resulting images have high fidelity to the class category.

We might expect that for a suitably chosen textual similarity threshold, the resulting dataset would bear resemblance to the original ImageNet. However, we demonstrate that this is anything but the case. The dataset, so created from LAION, very much looks unlike ImageNet. And we explain *why*, supported by

---

[1]Unless otherwise stated, by ImageNet we mean the ImageNet ILSVRC-2012 dataset.

independent evidence from other well-curated datasets. This explanation, although subtle, reveals a fundamental fact about the difference between ImageNet and LAION that has consequences for understanding dataset creation at large.

## 1.1 Our Contributions

We introduce a new research artifact, called *LAIONet*, that aims at a recreation of ImageNet on the basis of LAION. We start from LAION-400M, a collection of 400M image-text pairs extracted from web pages in Common Crawl (`commoncrawl.org`) between 2014 and 2021. The relevance of images and their corresponding texts was quality-controlled with OpenAI CLIP model, excluding instances with a cosine similarity of image and text embeddings less than 0.3.

**Creation of LAIONet.** We create LAIONet solely on the basis of text-based selection. We require the exact "lemmas" (terms) in a so-called "synset" of an ImageNet category to appear in the text corresponding to an image. Moreover, we require a high similarity between the text and the synset name and definition. We use the cosine similarity of CLIP text embeddings to calculate this similarity, however, we make consistent observations using MPNet (Song et al., 2020) as the text encoder. LAIONet selection criteria are conservative in that they tend toward images that are easy to classify; at least from the CLIP point of view, there is no evidence that LAIONet images are harder to classify than ImageNet.

**Contrasting LAIONet and ImageNet.** To begin to understand the differences between LAIONet and ImageNet, we evaluate a slew of Imagenet models on LAIONet. As we show, the accuracy of models trained on ImageNet drops by 5 to 12 percentage points when evaluated on LAIONet (Figure 1). In calculating accuracy, we weight classes uniformly as is done in ImageNet. When classes are weighted based on the frequency of each class in LAIONet, accuracy drops by another 5 to 10 percentage points.

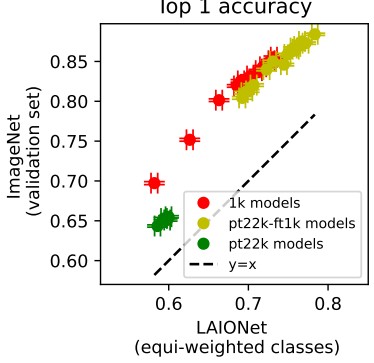 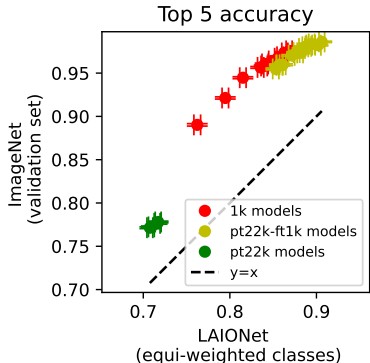

Figure 1: Accuracy of ImageNet-trained models when evaluated on ImageNet validation set versus LAIONet. Three types of models are distinguished based on whether they are pre-trained on ImageNet-22k and whether they are fine-tuned on ImageNet-1k. Accuracy is defined as the average of the recalls calculated for each class that is present in LAIONet.

Drops in accuracy, such as these, are a well-documented phenomenon in machine learning at this point. In this work, we go a step further by providing a substantive explanation for the difference between LAIONet and ImageNet.

**Diagnosing the Difference.** In the first step, we observe that the intra-class similarity, measured as the pairwise similarity of the images within a class, is lower for LAIONet than for ImageNet. In other words, LAIONet images are more diverse in each class. The recall of the models is also lower in the classes with lower intra-class similarity. Hence, lower intra-class similarity gives a concrete reason for why the accuracy of ImageNet models drops on LAIONet. But why does LAIONet have lower intra-class similarity in the first place?

We answer this question in terms of two plausible causal graphs for the respective data-generating processes (Figure 2). Both graphs are based on the standard anti-causal representation of classification problems (Schölkopf et al., 2012), whereby for each category $Y$ there is a mechanism to generate data (here, image $X$ and text $T$) given $Y$. But, the graphs differ in one important aspect.

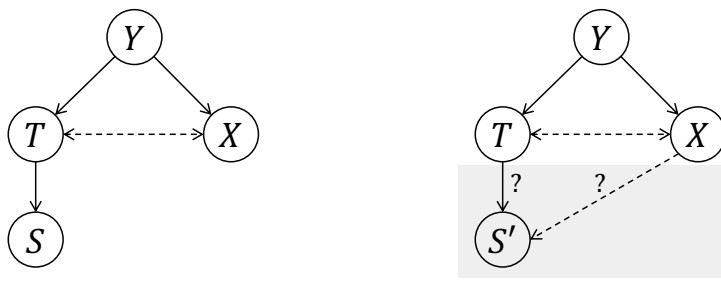

(a) LAIONet data collection          (b) ImageNet data collection

Figure 2: The suggested underlying mechanism of data generation and selection in LAIONet and ImageNet. Class $Y$, text description $T$, image $X$, selection $S$ or $S'$. The dashed line between $X$ and $T$ means that such links can exist in the graph.

In the case of LAIONet (Figure 2a), selection is based on text alone. The causal graph has the important property that the distribution of the image is independent of the selection decision conditional on the text. In other words the text serves as an information bottleneck between the selection mechanism and the image. Choosing an image reveals nothing more about the image than what can be learned from its textual representation. This powerful conditional independence property limits how much selection can bias the distribution of the image. In contrast, in the case of ImageNet (Figure 2b), there is a link from the image to the selection decision. For example, this link exists when human annotators see the full image and decide to select or discard an image. The existence of this link is what can strongly bias the distribution of the image conditional on selection. It is this selection bias that is visible in the higher intra-class similarity.

Our case hinges on the existence and strength of the image-to-selection link in the causal graph for ImageNet. We then go beyond LAIONet and provide three complementary arguments as evidence:

- We can weaken the image-to-selection link by considering ImageNet instances of different *selection frequencies.* The selection frequency describes the rate at which Amazon MTurk workers selected a candidate image into the dataset within a target class. This allows us to modulate the strength of the image-to-selection link. Looking at three versions of ImageNetV2 (Recht et al., 2019), we find that for a lower selection frequency, the resulting images come closer to LAIONet.

- We show that text alone cannot explain why an image was selected into ImageNet. The ImageNet-Captions dataset (Fang et al., 2022) has restored the captions for one-third of the original ImageNet images. If the text was the only factor in determining the relevance to a synset, it should explain why the images in ImageNet-Captions are selected. Looking at the similarity between texts and their synsets, a majority of text-synset pairs exhibit high similarity, but the distribution has a heavy tail and there are instances with low similarity. For pairs with low similarity, there are often many other synsets more similar to the text. This makes these instances unlikely to have been selected solely based on their text.

- We search LAION for the texts most similar to the texts from the ImageNet-Captions dataset. The resulting images show significantly higher variability (in other words, lower intra-class similarity) than ImageNet. This suggests that another mechanism must have been at play.

In conclusion, we argue that the image-to-selection mechanism was significantly at play in the creation of ImageNet. It is this mechanism that makes ImageNet look unlike LAION. This insight has direct prescriptive

value for dataset creation efforts in general. When creating a dataset and diversity is desired, we should select candidates on the basis of an information bottleneck. A succinct text caption, for example, generally carries much less information than the entire image. Selecting on the basis of the text caption, therefore, retains much of the entropy present in the image distribution.

All code is available at: `https://github.com/alishiraliGit/eval-on-laion`

## 1.2 Related Work

Torralba & Efros (2011) introduced cross-dataset evaluation in computer vision and specifically illustrated the uniqueness that each common dataset at the time, including ImageNet, possesses. Recreating an ImageNet test set, called ImageNetV2, although with a different motivation, was the subject of the seminal paper by Recht, Roelofs, Schmidt, and Shankar (2019). Engstrom et al. (2020) argue that there is a subtlety in thresholding empirical estimates of the true underlying selection frequency of an image in ImageNetV2. Our argument, however, does not rely on any specific threshold of the selection frequency. We only need to observe what happens as we vary it from small to large. In contrast to ImageNetV2, our goal is not to recreate ImageNet as closely as possible. Rather it is the differences between ImageNet and LAION that are the focus of our investigation.

Many other works have modified ImageNet for a variety of reasons. Geirhos et al. (2019) created a stylized version of ImageNet to reduce the reliance of the trained model on texture. Xiao et al. (2021) disentangled the foreground and background of ImageNet images to show the tendency of the models to rely on the background. Li et al. (2023b) proposed ImageNet-W test set by inserting a transparent watermark into the images of ImageNet validation set, revealing the reliance of the models on watermarks. ImageNet undergoes ongoing augmentation over time. For example, the ImageNet-Captions (Fang et al., 2022) project has restored the captions of about one-third of original ImageNet images from Flickr. ImageNet-X (Idrissi et al., 2023) provides a set of human annotations pinpointing 16 failure types for ImageNet such as pose, background, or lighting. The peculiarities of ImageNet have been the subject of multiple studies. For example, Huh et al. (2016) found the large size and many classes, including very similar classes, do not affect the successful transfer performance of ImageNet-trained features.

On the side of LAION, researchers are keenly interested in understanding the strong zero-shot accuracy of contrastive language image models using LAION (Vogel et al., 2022). Fang et al. (2022) found none of the large training set size, language supervision, and contrastive loss function determines this robustness and a more diverse training distribution should be the main cause. Our work demystifies this distributional advantage by contrasting ImageNet and LAION. Nguyen et al. (2022) compared various large image-text datasets differing in the creation process and found the robustness induced by each varies widely in different aspects, suggesting further studies of the role of dataset design. Our work highlights an important mechanism at play in dataset design that can move the dataset further away from a natural distribution.

## 2 LAIONet: An ImageNet Out of LAION

Our starting point is to create an ImageNet-like dataset from LAION. This dataset is a research artifact intended to highlight the differences between LAION and ImageNet. Our goal is not to provide a new benchmark or a new training set. However, LAIONet might be of interest to obtain diverse samples, or variants of LAIONet may be created to improve our understanding of benchmarks.

To start, recall that every ImageNet class corresponds to a WordNet (Miller, 1998) *synset* which consists of so-called *lemmas*. Synsets also come with a short definition known as gloss. We label a LAION instance with a WordNet synset if 1) at least one lemma from the synset exists in the text of the instance, and 2) this text is sufficiently similar to the name and definition of the synset. Out of LAION 400M samples, 21M of them passed the first condition. The second condition ensures the lemma as found in the LAION sample has the intended meaning. To quantify the similarity of the LAION text and a synset, we first create a textual representation for the synset by concatenating its name and definition (to be called the synset text). We then calculate the embedding vectors for both the synset text and LAION text using CLIP and compute

their cosine similarity. Alternatively, one may use any sufficiently powerful text encoder for this purpose. For instance, we repeat this process using MPNet (Song et al., 2020) in Appendix A.

Figure 3a illustrates the distribution of LAION text to synset text similarities. In general, a high value for textual similarity ensures the LAION text is describing the same object as the synset. But as Figure 3b shows, we cannot set a very high similarity threshold since the extracted dataset will lose its coverage over the ImageNet's 1k classes. We found the threshold of 0.82 the highest reasonable choice as it allows for covering most classes while going beyond it sharply reduces the number of covered classes (Figure 3b) with no significant reduction in the dataset size (Figure 3c). To further support this choice, in Section 4 (Figure 11b), we demonstrate that using the restored captions of ImageNet, a textual similarity of above 0.7 is sufficient to ensure that a sample belongs uniquely to the synset. Also refer to Appendix C for an example of when the second step of filtering is necessary and why the chosen threshold is conservative.

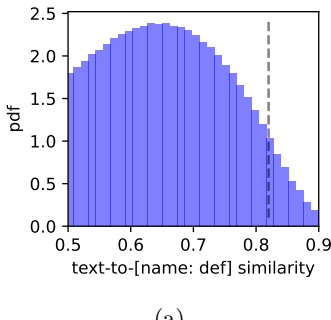 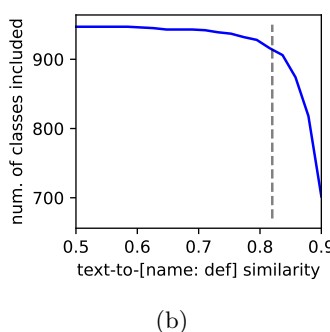 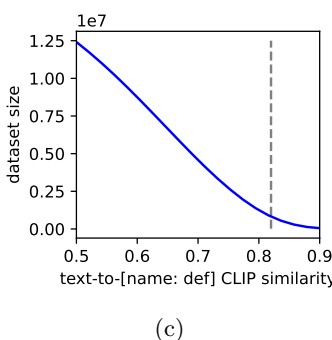

|(a)|(b)|(c)|

Figure 3: Filtering LAION samples based on their textual similarity to the candidate synsets. The dashed line shows the chosen threshold. (a) The overall probability density function (pdf) of the similarities prior to the second step of filtering. (b and c) The number of ImageNet classes covered by at least one example in the dataset and the size of the dataset for different levels of similarity threshold.

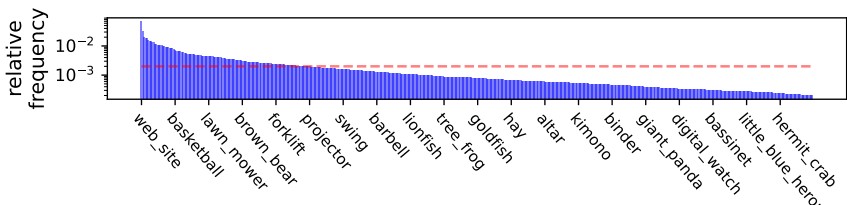

Figure 4: Relative frequencies of different classes in LAIONet sorted in descending order for the 500 most frequent classes. Some class names are shown. The red line shows uniform weight.

We take a few additional measures to guarantee the safety and quality of the chosen instances. First, we drop samples with more than one label to simplify the evaluation on the dataset. Second, we drop images tagged as not-safe-for-work in LAION. Finally, we exclude images that contain text matching the name of their synset. This will ensure the captions are describing an object in the image and not just reflecting on another text. To achieve this, we employ EAST for text detection (Zhou et al., 2017) and TrOCR for text recognition (Li et al., 2023a). This step eliminates 1.1% of the samples.

The final dataset, which we call *LAIONet*, consists of 822k samples from 915 ImageNet classes, sufficiently large for fine-grained evaluation purposes at statistical significance. Unlike ImageNet which provides about the same number of images per class, the large variation in the relative frequency of the classes in LAIONet reflects the natural distribution of each class (Figure 4). In particular, 95% of LAIONet classes have at least 6 samples, 90% have at least 13 samples, 80% have at least 36 samples, and 70% have at least 75 samples. We will later use the relative frequency of classes to compare the performance of models in frequent and infrequent classes. Note that we can also create a more conservative version of LAIONet mimicking the ImageNet validation set by retaining only the top 50 most similar instances for each class. This version

of LAIONet yields consistent observations in general (Appendix B). Find sample images of LAIONet in Appendix I.

Are LAIONet images harder to classify? To find out, we compare CLIP zero-shot accuracy on LAIONet and ImageNet. For every image, we predict the label of the image based on what synset has the highest cosine similarity between the image embedding and the synset text embedding. To make accuracy estimates on LAIONet comparable with ImageNet, we calculate accuracy as the average recall across the classes present in LAIONet. This uniform weighting is consistent with the setup of ImageNet validation with 50 images per class. We found CLIP zero-shot top 1 accuracy to only differ by 2% across datasets. Hence, at least from the CLIP view, LAIONet images are not harder to classify in terms of average accuracy. We acknowledge a limitation in that the CLIP text embeddings used for LAIONet creation are jointly trained with the image embeddings used for zero-shot accuracy calculations. This may give CLIP an advantage on LAIONet, and CLIP zero-shot accuracy should be interpreted with caution. Appendix D offers a more direct assessment of the level of difficulty involved in identifying the intended object in LAIONet. This is achieved by directly computing the cross-modality similarity between an image and its associated synset. Overall, LAIONet images do not exhibit significant difficulty compared to ImageNet.

## 3 LAIONet Versus ImageNet

We begin to understand the differences between the two datasets by looking at the accuracy of various ImageNet classifiers on LAIONet. After observing a significant accuracy drop, we consider the disparity in intra-class similarity as a possible explanation.

### 3.1 Comparing Accuracy

We consider four model families: ResNet (He et al., 2016), Vision Transformers (ViT) (Dosovitskiy et al., 2021), modernized ConvNet (ConvNeXt) (Liu et al., 2022), and Bidirectional Encoder representation from Image Transformers (BEiT) (Bao et al., 2022). All models are trained on ImageNet without extra training data. We use various versions of each model in terms of the size (small, base, large, etc.), image resolution (224x224 or 384x384), patch resolution (16x16 or 32x32), and whether models are pre-trained on the complete ImageNet with 22k classes or not. All models come from HuggingFace (`huggingface.co`) checkpoints.

We first compare the (equally weighted) accuracy defined by the average of recalls across the classes covered by LAIONet. Figure 1 compares the top 1 and top 5 accuracy on ImageNet and LAIONet. In most of the highly accurate models, accuracy drops by at least 10 percentage points when estimated on LAIONet with models pre-trained on ImageNet-22k showing slightly more robustness.

Next, we use the relative frequency of each class in LAIONet to weight its recall and obtain a LAION-weighted accuracy. Figure 5 compares LAION-weighted and equally-weighted accuracy on LAIONet. The LAION-weighted accuracy is consistently lower by 5 to 10 percentage points. This can partially be explained by the observation that ImageNet-trained models are performing worse when the class is describing a more common object (Appendix G.1). We also compared LAION-weighted and equally-weighted accuracy on ImageNet and found consistent results in Appendix F.

### 3.2 Comparing Intra-Class Similarity

While LAIONet images are in a precise sense not more difficult than ImageNet, there is another factor that can explain the accuracy drop: the intra-class similarity of images. We define this similarity as the pairwise similarity of the images from the same class, measured by the cosine similarity of their CLIP image embeddings. The lower these similarity values, the more diverse the images from that class.

Figure 6a shows the distribution of intra-class similarities aggregated over all the classes that have at least 7 samples in LAIONet. To make the distributions comparable, we sampled (with replacement) the similarities from LAIONet to match ImageNet. The left tail of the LAIONet intra-class similarity distribution makes it clear that LAIONet overall provides a more diverse set of images. To observe the effect in greater detail, for each class, Figure 6b shows the average intra-class similarity of LAIONet images subtracted by the average

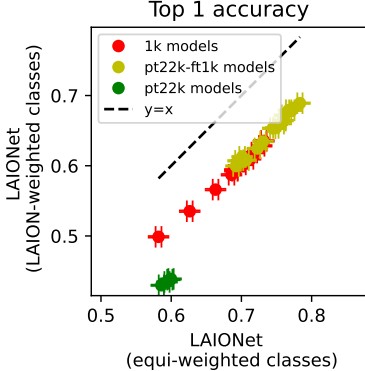 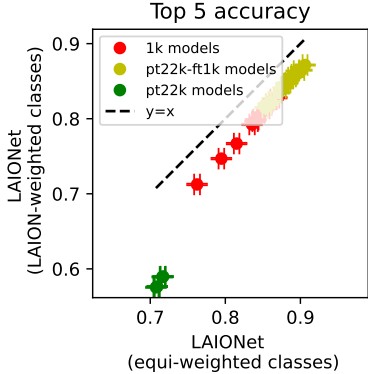

Figure 5: A LAION-weighted accuracy is calculated according to the relative frequency of the classes in LAIONet and compared to the accuracy with equally weighted classes.

intra-class similarity of ImageNet images from the same class. In almost two-thirds of the classes, LAIONet has significantly lower intra-class similarity. This provides further evidence that LAIONet images exhibit greater variability within each class.

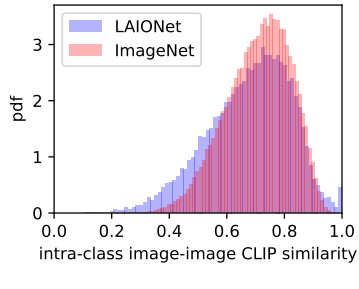

(a) Dist. of aggregated similarities

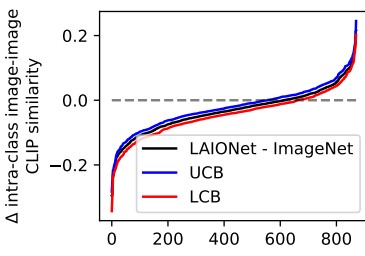

(b) Comparison across classes

Figure 6: Comparing the intra-class similarity of LAIONet and ImageNet. (a) In each class, pairwise similarities of LAIONet images are sampled to match ImageNet in number. All the classes combined, the distribution of intra-class similarity is depicted. (b) For each class, the average intra-class similarity of ImageNet images is subtracted from the same value in LAIONet. The blue and red curves show upper and lower 95% confidence intervals. All values are sorted ascendingly.

In Appendix G.2, we show that models struggle more with classes where LAIONet and ImageNet have significantly different intra-class similarity. This, combined with our observation of LAIONet having lower intra-class similarity, supports our argument that intra-class similarity plays a crucial role in reducing accuracy.

## 4   Diagnosing ImageNet

As is standard modeling practice, we think of a data-generating process that for a given class $Y$ generates a pair of image $X$ and text $T$. Ideally, when we search for images of a particular class $y$, we would like to draw samples from distribution $p(X|Y = y)$. Unless we have access to the generative process or we have a completely random set of images all correctly labeled, drawing samples directly from $p(X|Y = y)$ will not be possible. In particular, none of these options are available when researchers collect a new dataset. Instead, researchers have to define a selection mechanism $S$ for choosing images. What we observe is the conditional distribution of $X$ given $S$.

In creating LAIONet, we relied on texts to select the samples (Figure 2a). LAIONet images follow $p(X|S = 1)$, where $S = 1$ if $T$ is sufficiently similar to $Y$. With our conservative selection criteria, we can assume every $T$ passed our similarity threshold is generated from the intended $Y = y$. Therefore, $p(X|S = 1) = p(X|S = 1, Y = y)$. Generally, an image carries much more information than the text. So, for the images of a certain class, conditioning on the text alone should not alter the distribution significantly. Intuitively speaking, $p(X|Y = y, T = t) \approx p(X|Y = y)$. In our setting, a weaker independence is sufficient to show LAIONet images follow the desired distribution. Even if information from $X$ beyond $Y$ is present in $T$, since we deliberately refrained from searching for visual descriptions in the text, we expect $S$ to be independent from $X$ for a given $Y = y$. Hence, we have reason to hope $p(X|S = 1) \approx p(X|S = 1, Y = y) \approx p(X|Y = y)$.

In general, a selection $S'$ can rely on both text and image directly (Figure 2b). In this case, the distribution of observed images $p(X|S' = 1)$ can be far from the desired distribution $p(X|Y = y)$. We believe this has happened in the collection of ImageNet, primarily through human annotators examining and acting on images. Incorporation of visual features at the side of the search engine provider is another plausible mechanism. While we may not be able to pinpoint the exact mechanism at play, we will now move beyond LAIONet and demonstrate, through three independent experiments, a strong link between the image $X$ and the selection criterion $S'$ in the creation of ImageNet.

## 4.1 A Weaker Image-To-Selection Link Makes ImageNet More Like LAIONet

Image annotation is one clear mechanism by which the image $X$ influences selection $S'$. Changing the strictness of annotation allows us to modulate the strength of this mechanism and measure its effect. This experiment is possible due to the availability of ImageNetV2 (Recht et al., 2019) that comes with three different versions. The three versions of ImageNetV2, called a, b, and c, differ in the level of agreement among annotators. More precisely, each image comes with an *MTurk selection frequency* which is what fraction of MTurk workers selected the image to be from the target class. ImageNetV2 versions a, b, and c have an average MTurk selection frequency of 0.85, 0.73, and 0.93, respectively. Note that version b has the lowest and version c has the highest selection frequency.

We first observe that allowing for more disagreement among annotators results in the inclusion of more diverse images. Figure 7a shows the distribution of intra-class similarity for ImageNetV2 versions b and c. One can see that in version b, with the lowest average MTurk selection frequency, the intra-class similarity is shifted toward lower values.

We next further show as the average MTurk selection frequency increases, ImageNetV2 becomes more similar to ImageNet and less similar to LAIONet. In this regard, to compare two datasets, we count the number of classes in which the first dataset has significantly lower intra-class similarity than the second dataset, and vice versa. Figure 7b compares LAIONet and three versions of ImageNetV2. As the figure suggests, LAIONet and ImageNetV2 are similarly diverse when the average MTurk selection frequency is low (corresponding to ImageNetV2 version b). However, as the MTurk selection frequency increases, ImageNetV2 shows higher intra-class similarity than LAIONet. At the same time, Figure 7c shows ImageNetV2 becomes more similar to ImageNet as we increase the MTurk selection frequency.

We also contrast the accuracy of ImageNet-trained models on LAIONet and ImageNetV2 in Figure 8. Reaffirming our previous observation, as the MTurk selection frequency declines from version c to a to b, the accuracy on ImageNetV2 and LAIONet increasingly aligns. Notably, most models experience similar accuracy drops on ImageNetV2-b and LAIONet.

Together, these observations show the impact the image has on the selection, particularly during annotation, is significant and can partially explain the divergence between LAIONet and ImageNet. Further, the extra intra-class diversity of LAIONet is achievable from less stringent human annotation and can explain the consistent accuracy drop on LAIONet and ImageNetV2.

## 4.2 Introducing an Image-To-Selection Link Makes LAIONet More Like ImageNet

We deliberately refused to use visual information in selecting LAIONet samples. Complementing Section 4.1, a natural question arises: What would happen if we used multimodal LAION image-to-synset text similarity,

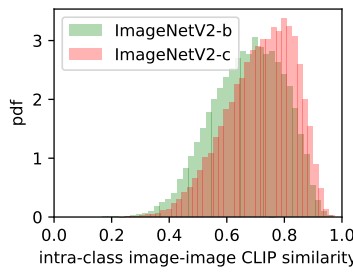 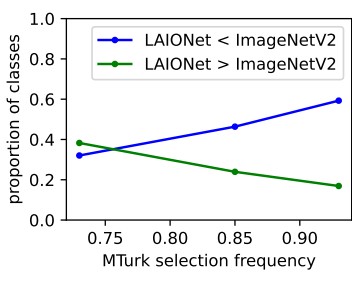 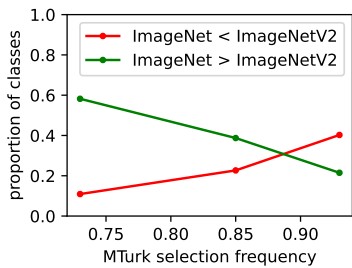

(a) Dist. of aggregated similarities  (b) LAIONet vs. ImageNetV2  (c) ImageNet vs. ImageNetV2

Figure 7: The effect of MTurk selection frequency on intra-class similarity. (a) The distribution of intra-class similarity aggregated over all classes for ImageNetV2 versions b and c. (b) LAIONet versus three versions of ImageNetV2. The vertical axis shows the proportion of classes in which one dataset has significantly lower intra-class similarity than the other. To determine whether intra-class similarity is significantly lower in a class, we compute the average intra-class similarity of images within that class for both datasets and compare their difference using a 95% bootstrapped confidence interval. Blue curve: the proportion of classes where LAIONet has lower intra-class similarity than a specific version of ImageNetV2. Green curve: ImageNetV2 has lower intra-class similarity. (c) ImageNet versus ImageNetV2. Red curve: ImageNet has lower intra-class similarity. Green curve: ImageNetV2 has lower intra-class similarity.

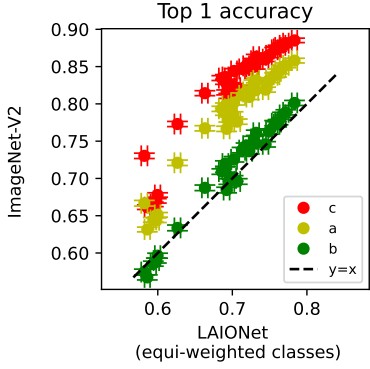 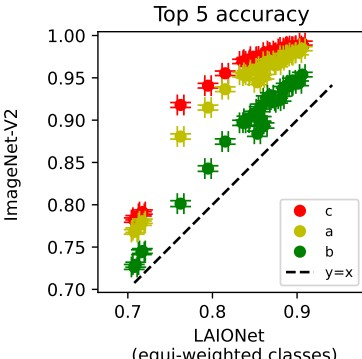

Figure 8: Accuracy of ImageNet-trained models evaluated on three versions of ImageNetV2 versus LAIONet.

either in place of or alongside LAION text-to-synset text similarity, to select samples? Answering this question, we construct new datasets out of LAION where we require an included sample to have image-to-[name: def] similarity greater than a threshold *and* text-to-[name: def] similarity also greater than another threshold. To choose these two thresholds, we control the total number of selected samples to be similar to our original version of LAIONet. Using CLIP embeddings to calculate similarities, we derive a multimodal similarity threshold for each value of the textual similarity threshold as shown in Figure 9. From this curve, we select four distinct threshold values and create a dataset for each. We call these datasets A, B, C, and D as depicted on Figure 9.

For all four datasets, we calculate the intra-class similarities of all the common classes among them. We show the distribution of intra-class similarities of datasets A, B, and D in Figure 10a. As the figure suggests, dataset A, which is the most similar to LAIONet, overall has the lowest intra-class similarities. As we strengthen the image-to-selection link by using a lower textual similarity and a higher multimodal similarity, datasets show higher intra-class similarity. This is also evident in Figure 10b where we plot the average of intra-class similarities across classes. In Figure 10c, we also look into the number of classes where a dataset has lower intra-class similarity compared to ImageNet. Figures 10b and 10c together show that dataset A, the most similar to LAIONet, exhibits greater diversity than ImageNet, both in terms of average intra-class similarity and the number of classes where A is more diverse. However, as we increase the multimodal

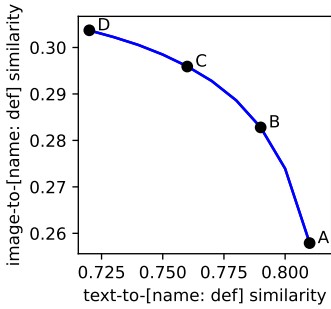

Figure 9: For each textual similarity threshold, we find the corresponding multimodal similarity threshold such that the resulting dataset, satisfying both similarity criteria, contains the same number of samples as LAIONet. We then choose four points on this curve, A, B, C, and D, and create four such datasets.

threshold, the datasets consistently become less diverse, eventually even less diverse than ImageNet. To make sure our measurement of intra-class similarities is not biased by the CLIP visual encoder, we have repeated this experiment using a variety of encoders in Appendix H and found even stronger results.

In summary, in this experiment, we used CLIP multimodal similarity to introduce an image-to-selection link into our dataset creation process. Our observations strongly suggest that this link can reduce the diversity of the included images when we control the total number of selected examples. While CLIP multimodal similarity, which is known to suffer from specific biases (Tong et al., 2024), may not accurately reflect how the original image-to-selection link worked in ImageNet, it demonstrates how relying on a richer modality can make the selection process more prone to bias.

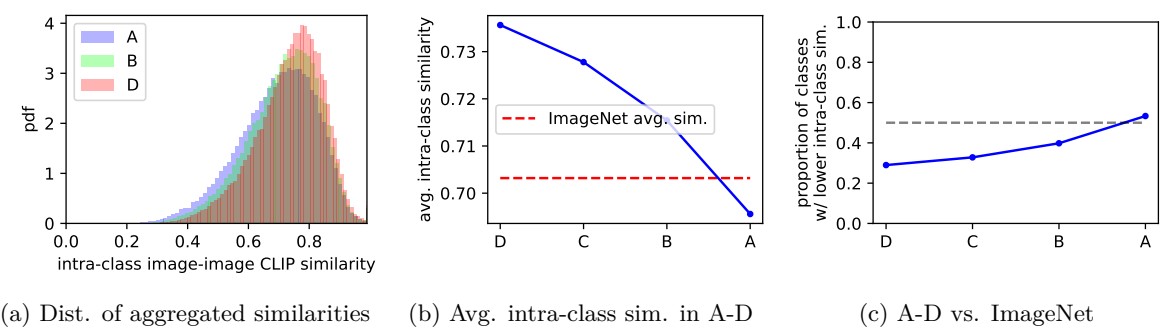

(a) Dist. of aggregated similarities     (b) Avg. intra-class sim. in A-D     (c) A-D vs. ImageNet

Figure 10: (a) The distribution of intra-class similarity aggregated over all classes for datasets A, B, and D. (b) The average of intra-class similarities compared across datasets A, B, C, and D. To calculate this average, we first find the average of intra-class similarity in each class and then take their average across all common classes of A, B, C, D. (c) The proportion of classes where the new datasets have lower-intra class similarities than ImageNet.

### 4.3 Text Alone Cannot Explain Why an Image Is Selected Into ImageNet

ImageNet-Captions (Fang et al., 2022) is a subset of ImageNet-1k training data with restored title, description, and tags from Flickr. We assume the samples in ImageNet-Captions are a random subset of the original ImageNet and the captions are accurately restored. If there was no link $X \to S'$, the accompanying caption of an image in ImageNet-Captions should be able to explain why this image is selected.

We follow Fang et al. (2022) and define the text as the title, description, and tags concatenated. Figure 11a illustrates the similarity between the texts and their respective synsets using CLIP text embeddings. Although most of the texts have a high similarity of 0.6 or above to their synsets, the distribution has a heavy left tail. The fact that a text has low similarity to the intended synset does not necessarily mean it could

not be chosen by the search engine. However, we show many of the texts that have low similarity to the intended synsets actually have high similarity to numerous other synsets, making them less likely to have appeared for the intended meaning. For every text, we find the similarity to all synsets, i.e. the similarity to their names and definitions, and count the proportion of unintended synsets (false classes) that are more similar to the text than the intended synset. A low value for this proportion shows the text well represents its intended synset whereas a significant non-zero value indicates that there are considerable other synsets that are more strongly present in the text. As Figure 11b demonstrates, for a text with low similarity to its synset there are on average 5% (equivalently, 200) or more other synsets more similar to the text. These observations show that at least based on the restored texts in ImageNet-Captions, the text alone cannot fully explain why an image is selected and another mechanism should have been at play.

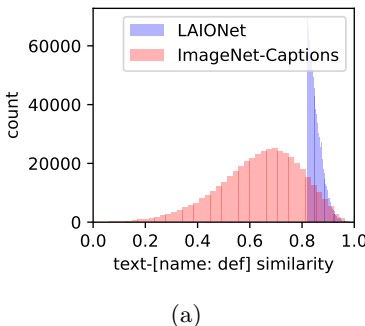 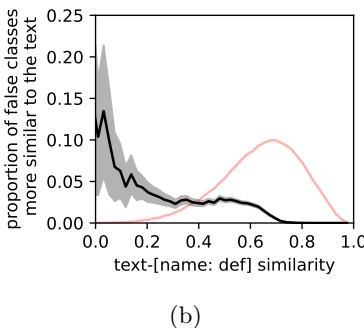

(a)                                (b)

Figure 11: (a) The distribution of the text-to-synset similarity. (b) For every bin of text-to-synset similarity, the average proportion of unintended classes which are more similar to the text than the intended class is depicted in black.

## 4.4 ImageNet, Had It Been Created Solely Searching Texts, Does Not Resemble Current ImageNet

If the link from $X$ to $S'$ did not exist, regardless of how the selection algorithms works, $p(X|T = t)$ would look similar in both graphs of Figure 2. To test this hypothesis, we extract a new dataset from LAION. For every image in ImageNet with corresponding text $T = t$ in ImageNet-Captions, we find the LAION sample with the most similar text to $t$. We only keep a LAION sample if the similarity is above 0.7. This choice ensures the two texts are sufficiently similar as we can consider them roughly the same while the dataset covers more than 95% of the ImageNet classes (Appendix E).

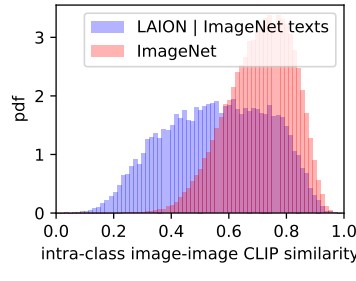 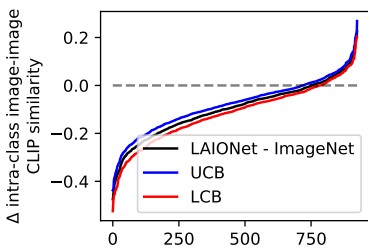

(a) Dist. of aggregated similarities                 (b) Comparison across classes

Figure 12: Comparing the intra-class similarity of the new dataset and ImageNet. The new dataset is obtained by selecting LAION examples with the most similar texts to the texts in ImageNet-Captions. (a) Distribution of intra-class similarity aggregated across all classes. In each class, pairwise similarities of the images in the new dataset are sampled to match ImageNet in number to make the distributions comparable. (b) For each class, the average of the intra-class similarity of the images in the new dataset minus the corresponding value in ImageNet is plotted in black. The upper and lower 95% confidence bounds are depicted in blue and red. All values are sorted ascendingly.

As Figure 12a suggests, images in the new dataset have a significantly lower intra-class similarity. Looking at each class separately, Figure 12b shows in almost 70% of the classes, the images from the new dataset are significantly more diverse (have lower intra-class similarity). These observations reject the hypothesis that the graphs of Figure 2 have the same structure and show a potential leak from the image to the selection. We note the limitation that texts in the ImageNet-Captions dataset may not completely include the text available at the time of ImageNet creation. Second, for many cases, we were unable to find great matches for the ImageNet texts in LAION-400M and scaling our analysis to LAION-5B might help here.

## 5 Conclusion

In conclusion, we argue that the image-to-selection mechanism played a significant role in the creation of ImageNet, distinguishing it from LAION. We demonstrated this through three experiments. First, we modulated the speculated link from image to selection, showing the significant contribution this mechanism has in reducing the diversity of the selected images. The next two experiments rejected the hypothesis that image plays no or negligible role in the selection by showing ImageNet captions cannot solely explain the selection.

This insight carries valuable implications for dataset creation efforts in general. When developing a new dataset and diversity is desired, we advise selecting candidate instances based on an information bottleneck, like a succinct textual description of the instance, rather than the full instance. This will mitigate the selection bias that may otherwise distort the distribution of data conditional on selection.

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

## A   An MPNet-Filtered LAIONet

The creation of LAIONet relies on textual similarity of the LAION text and synset text. In Section 2 we used the cosine similarity of CLIP text embeddings to calculate this similarity, however, any sufficiently strong text encoder can be used for this purpose. In particular, we use MPNet (Song et al., 2020) fine-tuned on 1B sentence pairs with a contrastive objective by HuggingFace.[2] We follow a similar procedure to Section 2 and choose the maximum similarity threshold so that the resulting dataset does not lose its coverage over classes. We select the similarity threshold of 0.58. As Figure 13 suggests, a threshold larger than 0.58 may exclude many classes without reducing the size of the resulting dataset. Refer to Appendix C for additional evidence that this threshold works.

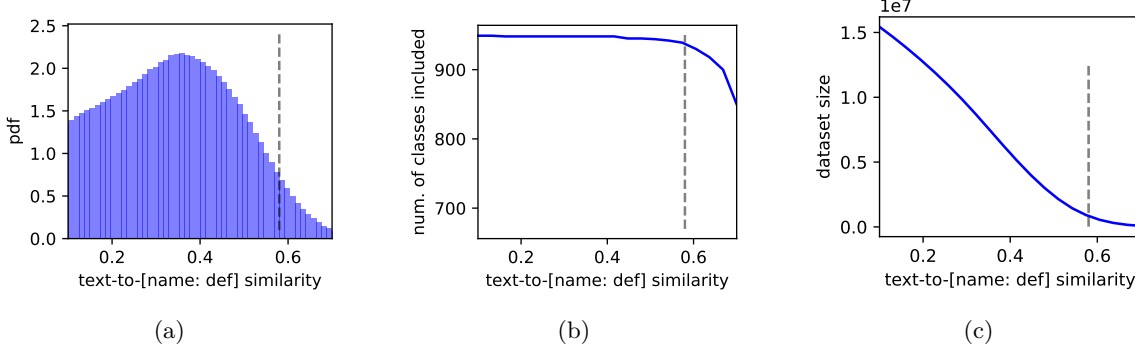

Figure 13: Filtering LAION samples based on their MPNet textual similarity to the candidate synsets. The dashed line shows the chosen threshold. (a) The overall distribution of the similarities prior to the second step of filtering. (b and c) The number of ImageNet classes covered by the dataset and the size of the dataset for different levels of similarity threshold.

Proceeding with the similarity threshold of 0.58, and after dropping samples labeled as not-safe-for-work, samples with multiple labels, and images containing text of their associated synsets, this version of LAIONet will have 831k samples covering 938 classes. As Figure 14 shows, consistent with our observation from CLIP-filtered LAIONet, models trained on ImageNet experience 10 to 15 percentage points of accuracy drop on MPNet-filtered LAIONet.

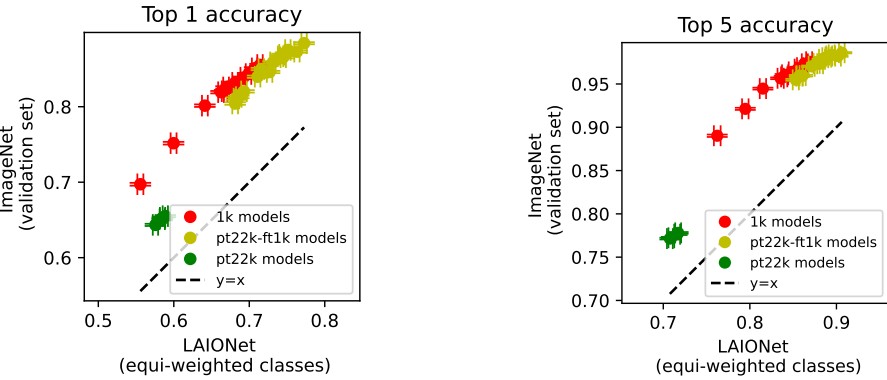

Figure 14: Accuracy of ImageNet-trained models when evaluated on ImageNet validation set versus MPNet-filtered LAIONet. Three types of models are distinguished based on whether they are pre-trained on ImageNet-22k and whether they are fine-tuned on ImageNet-1k. Accuracy is defined as the average of the recalls calculated for each class that is present in LAIONet.

---

[2]https://huggingface.co/sentence-transformers/all-mpnet-base-v2

Last but not least, Figure 15 suggests that MPNet-filtered LAIONet also exhibits lower intra-class similarity compared to ImageNet. In particular, in more than 70% of the classes, LAIONet has significantly lower intra-class similarity than ImageNet.

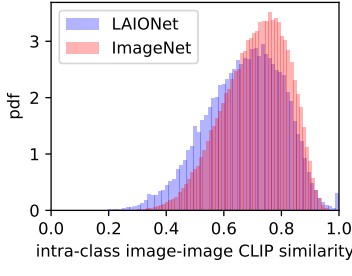

(a) Dist. of aggregated similarities

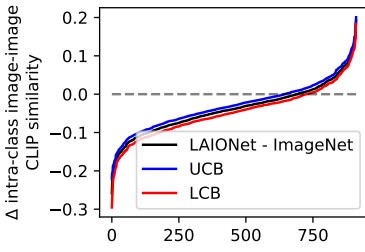

(b) Comparison across classes

Figure 15: Comparing the intra-class similarity of LAIONet and ImageNet. (a) In each class, pairwise similarities of LAIONet images are sampled to match ImageNet in number. All the classes combined, the distribution of intra-class similarity is depicted. (b) For each class, the average intra-class similarity of ImageNet images is subtracted from the same value in LAIONet. The blue and red curves show upper and lower 95% confidence intervals. All values are sorted ascendingly.

## B   A LAIONet From Most Similars

We created LAIONet by ensuring the presence of at least one lemma from the associated synset in the LAION text and by ensuring sufficient similarity between the synset text and LAION text. The frequency of each class in LAIONet reflects the natural distribution of that class on the web and likely worldwide. However, we can create a more conservative version of LAIONet by retaining only the top 50 most similar instances for each class. This will make LAIONet more similar to the ImageNet validation set. Such a version of LAIONet will have 39k samples covering 915 classes if initially filtered by CLIP similarity threshold of 0.82, and 41k samples covering 938 classes if initially filtered by MPNet similarity of 0.58.

Figure 16 illustrates that models performing well on ImageNet consistently experience a 7 to 10 reduction in accuracy on this version of LAIONet. Hence, the reduction in accuracy is consistent across all versions of LAIONets, including the most conservatively created ones. Figure 17 also confirms that this version of LAIONet still exhibits a longer tail of small intra-class similarity compared to ImageNet, potentially explaining the accuracy drop.

## C   On the Choice of LAION Text to Synset Text Similarity Threshold

In Section 2, we described how LAIONet is generated through substring matching LAION texts with ImageNet synset lemmas, followed by filtering out the cases where the LAION text is not sufficiently similar to the synset name and definition. A critical choice in the second filtering step is the choice of the minimum required textual similarity. We conservatively chose this threshold to be the largest value such that the remaining examples cover a large number of ImageNet's classes. To show this filtering is necessary and our threshold of 0.82 for CLIP-based filtering and threshold of 0.58 for MPNet-based filtering is conservative, we have provided an example in Figure 18. Here the synset "cougar" has lemma "puma". From WordNet definition, "cougar" is a "large American feline resembling lion". But the common usage of "puma" on the web is about a brand. As Figure 18 shows for small similarity to the synset, data most likely will represent the brand instead of the animal. As we increase the similarity threshold, the examples become more and more likely to be from the intended meaning. Our manual inspections show similar to this example, the chosen thresholds most likely result in high-quality matching to the intended meaning of the synset even if the web is dominated by other meanings.

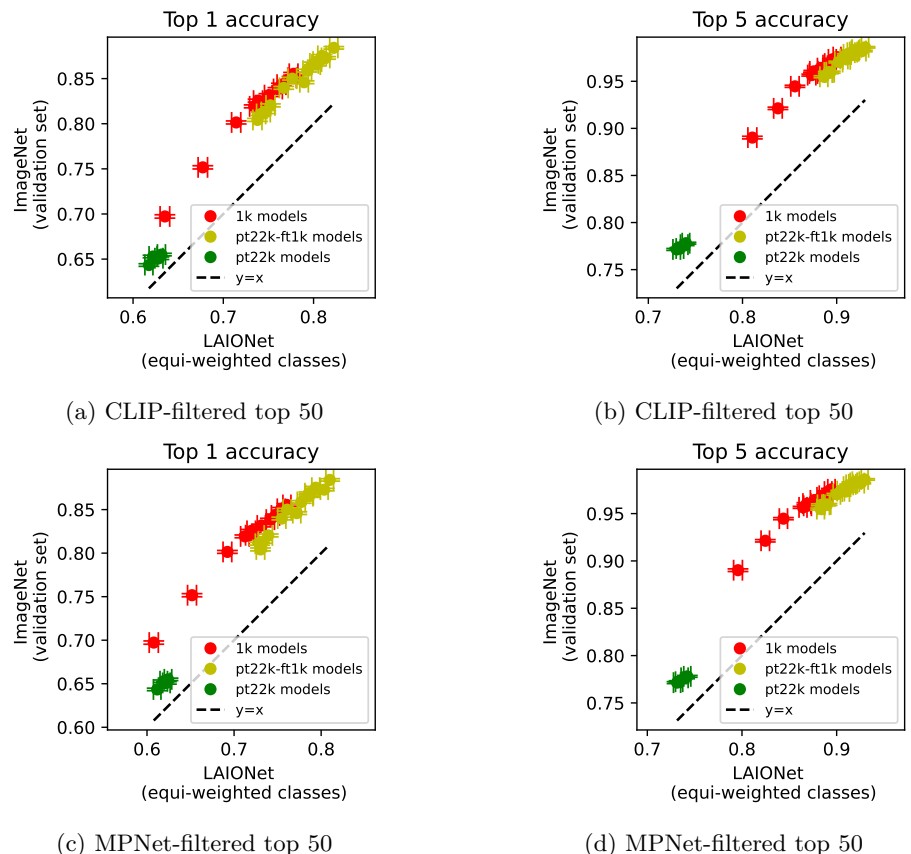

(a) CLIP-filtered top 50        (b) CLIP-filtered top 50

(c) MPNet-filtered top 50        (d) MPNet-filtered top 50

Figure 16: Accuracy of ImageNet-trained models when evaluated on ImageNet validation set versus LAIONet created by retaining top 50 most similar instances for each class.

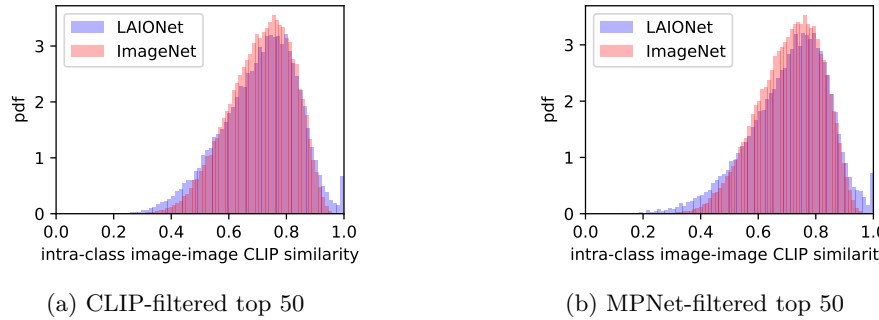

(a) CLIP-filtered top 50        (b) MPNet-filtered top 50

Figure 17: Comparing the intra-class similarity of LAIONet and ImageNet. In each class, pairwise similarities of LAIONet images are sampled to match ImageNet in number. All the classes combined, the distribution of intra-class similarity is depicted. LAIONet is created by retaining top 50 most similar instances to the synset text in each class after a textual similarity filtering with CLIP or MPNet.

# D  On the (Non)Difficulty of LAIONet Image Classification

To obtain a better idea of how hard is it to recognize an object in LAIONet, we calculate the cross-modal similarity of the images to the texts of their associated synsets using CLIP embeddings. A high value of image-to-synset similarity indicates CLIP is able to identify an object from the synset in the image. On the other hand, a low value could indicate that the intended object is either absent from the image or difficult

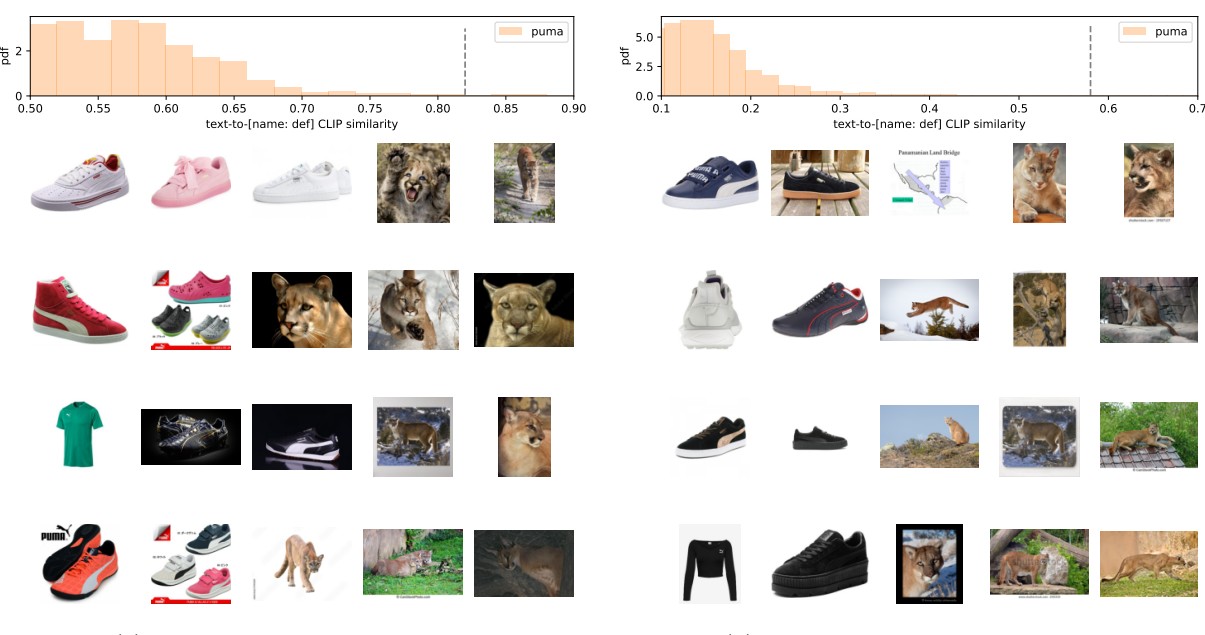

(a) CLIP-based textual similarity

(b) MPNet-based textual similarity

Figure 18: Sample images from five intervals of LAION text to synset text similarity.

to recognize. We compare the image-to-synset similarities obtained from the ImageNet validation set and LAIONet.

Figure 19a illustrates the distribution of image-to-synset similarity for LAIONet and ImageNet. To ensure these distributions are comparable, we sampled LAIONet with replacement to match the number of images per class in the ImageNet validation set. As the figure suggests, the two datasets are not significantly different. In a more fine-grained test, we compared the image-to-synset similarity of the LAIONet and ImageNet for each class. Figure 19b shows the average similarity in each class for LAIONet subtracted by the average similarity in the same class for ImageNet along 95% upper and lower confidence bounds. Overall, there is no strong signal that LAIONet images are harder in particular.

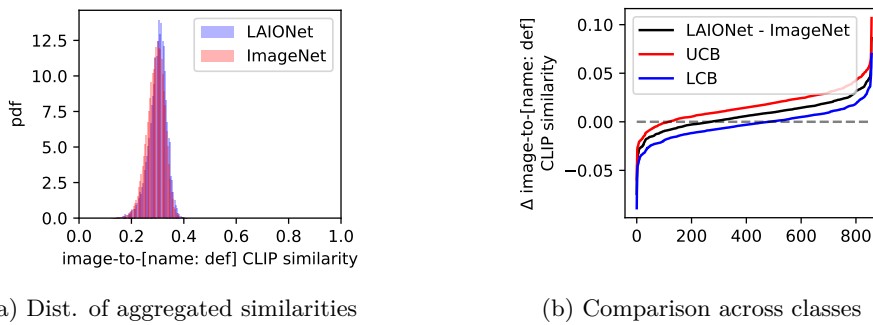

(a) Dist. of aggregated similarities

(b) Comparison across classes

Figure 19: Comparing image-to-synset similarities of LAIONet and ImageNet. (a) For each class, LAIONet is sampled with replacement to have the same number of images as ImageNet, and all samples are aggregated to obtain the distribution. (b) For every class, the average similarity of the images to synset text is calculated for LAIONet and ImageNet and the difference is plotted. The upper and lower 0.95% confidence bound for this difference is plotted in red and blue. All values are sorted ascendingly.

# E   On the Choice of Textual Similarity Threshold in Extracting Most Similar LAION Instances to ImageNet-Captions

In Section 4.4, we selected a similarity threshold of 0.7 as the minimum requirement for similarity between LAION text and ImageNet text in order to include a sample from LAION. Ideally, we look for LAION examples with identical text as the ImageNet but due to the limited number of samples available in LAION, this is not possible. As Figure 3b shows, increasing the similarity threshold beyond the chosen level of 0.7 significantly decreases the number of covered classes. Meanwhile, for larger thresholds, the new dataset looks more like ImageNet but is still distinguishable. As Figure 20b shows, the proportion of classes with significantly lower intra-class similarity in ImageNet increases as the threshold increases, while the proportion of classes with significantly lower intra-class similarity in the new dataset decreases. The gap still persists but can potentially become smaller in the region our data cannot cover. In sum, the new dataset extracted based on ImageNet looks unlike ImageNet but to the extent it is possible to find similar texts in LAION.

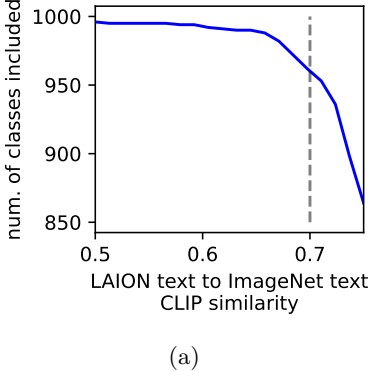
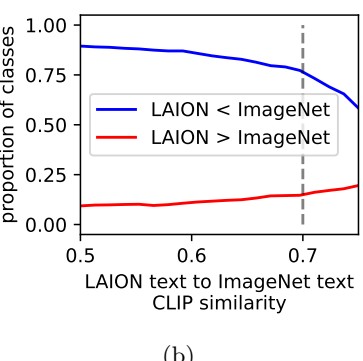

(a) (b)

Figure 20: The effect of similarity threshold on the dataset extracted from LAION samples with most similar texts to the ImageNet texts. (a) Number of the classes covered in the new dataset versus the similarity threshold. (b) Proportion of classes with significantly lower intra-class similarity in the new dataset (blue) and proportion of classes with significantly lower intra-class similarity in ImageNet (red) versus the similarity threshold.

# F   LAION-Weighted Versus Equally-Weighted Accuracy Evaluated on ImageNet

In Section 3.1 we introduced LAION-weighted accuracy where we use the relative frequency of each class in LAIONet to weight its recall. As we presented in Figure 5, the LAION-weighted accuracy is consistently lower than the equally-weighted accuracy when models are evaluated on LAIONet. This observation is not limited to evaluation on LAIONet. In fact, Figure 21 shows when we weight the classes according to their relative frequency on LAIONet, ImageNet accuracy also decreases. This can be attributed to the challenge of recognizing more frequent objects, given their potentially diverse types.

# G   The Relation of Recall, Relative Frequency, and Intra-Class Similarity

## G.1   Recall Versus Relative Frequency

In Section 3.1 we observed accuracy drops when we weight different classes according to their frequency in LAIONet. This can be partially explained as models perform worse in more frequent classes. To directly observe this, Figure 22 shows the recall in each class versus the relative frequency of the class in LAIONet. Regardless of whether LAIONet is created by filtering based on CLIP textual similarity or MPNet similarity, there exists a weak but consistent trend that more frequent classes are more likely to be misclassified.

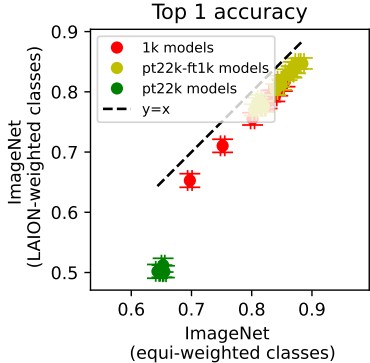 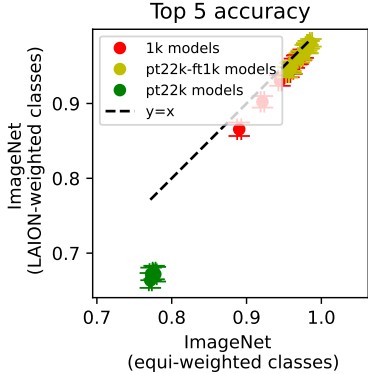

Figure 21: On ImageNet, a LAION-weighted accuracy is calculated according to the relative frequency of the classes in LAIONet and compared to the accuracy with equally weighted classes.

## G.2  Recall Versus Intra-Class Similarity

Section 3.2 introduced the hypothesis that higher intra-class similarity may account for the lower-than-expected performance of ImageNet models on LAIONet. To observe that intra-class similarity can be responsible for accuracy drop, Figure 23 demonstrates that models struggle on classes where LAIONet is more diverse than ImageNet, as shown by the recall rates plotted against the difference in average intra-class similarity. This is true regardless of what notion of accuracy and what version of LAIONet, CLIP-filtered or MPNet-filtered, is used.

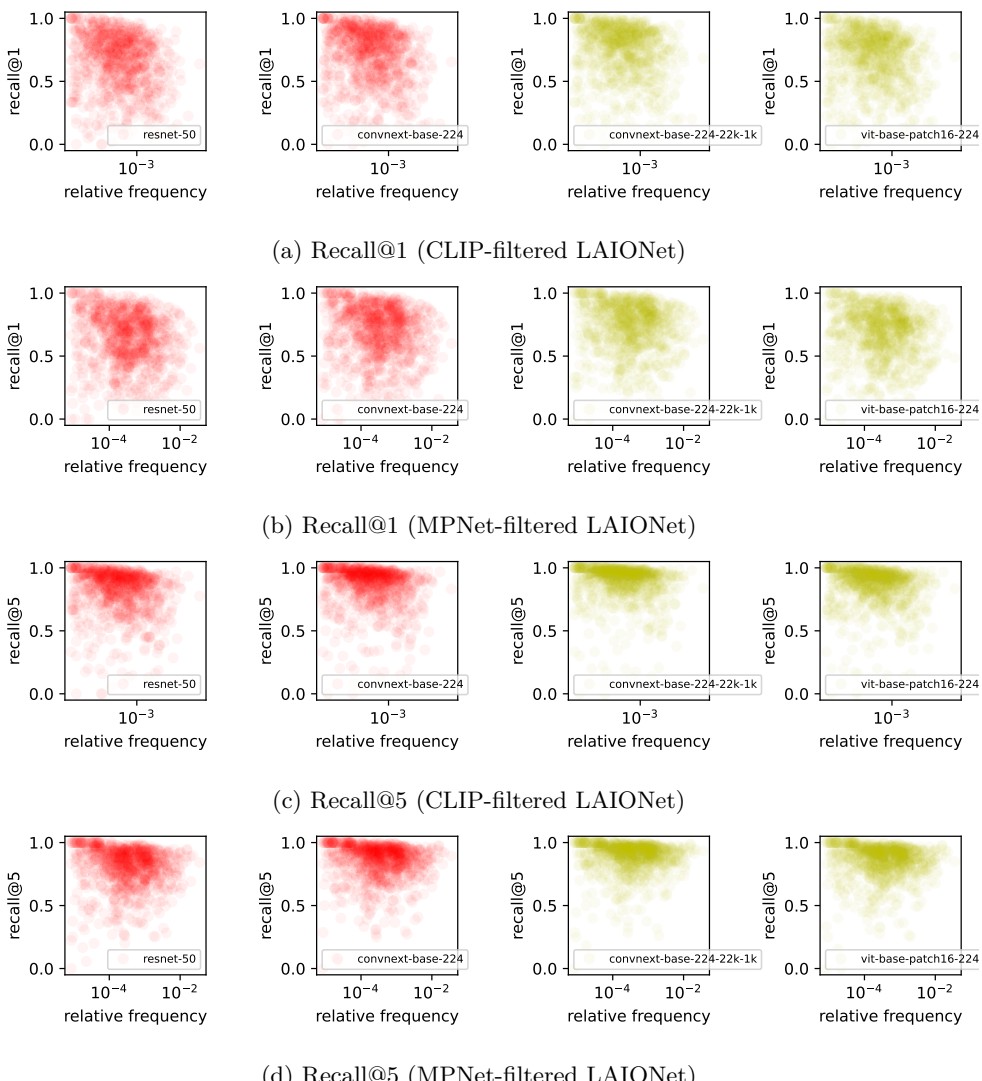

(a) Recall@1 (CLIP-filtered LAIONet)

(b) Recall@1 (MPNet-filtered LAIONet)

(c) Recall@5 (CLIP-filtered LAIONet)

(d) Recall@5 (MPNet-filtered LAIONet)

Figure 22: Recall per class evaluated on LAIONet versus how frequent the class is in LAIONet. Four different models are used, where two of them are pretrained on ImageNet-21k and two of them are not. Two versions of LAIONet, CLIP-filtered and MPNet-filtered are included. Trends are consistent.

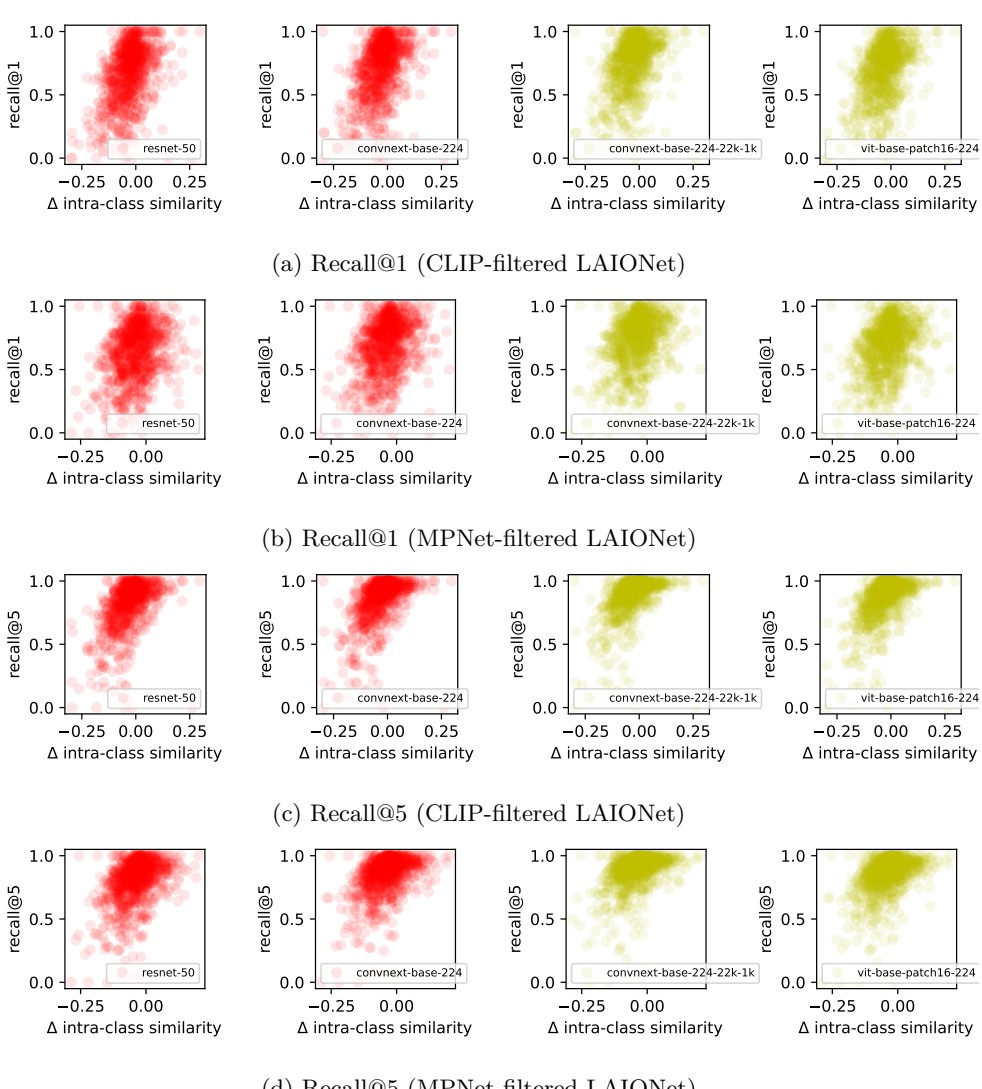

(a) Recall@1 (CLIP-filtered LAIONet)

(b) Recall@1 (MPNet-filtered LAIONet)

(c) Recall@5 (CLIP-filtered LAIONet)

(d) Recall@5 (MPNet-filtered LAIONet)

Figure 23: Recall on LAIONet for each class versus the disparity in intra-class similarity between LAIONet and ImageNet. This disparity (horizontal axis) is measured by subtracting the class-average intra-class similarity in ImageNet from that in LAIONet. Four exemplary models are shown, where two of them are pretrained on ImageNet-21k (yellow) and two of them are not (red). Two versions of LAIONet are considered. Trends are consistent.

# H  Using a Variety of Image Encoders to Calculate Intra-Class Similarities

In Section 3.2, we introduced intra-class similarity and used CLIP image embeddings to calculate that. Since CLIP has well-known visual biases (Tong et al., 2024), we repeat our calculations using a variety of other encoders. This includes using the last hidden layer of the base ViT, BEiT, and ConvNeXT models, all pre-trained on the larger ImageNet-22k datasets.

Figure 24 illustrates the proportion of classes where LAIONet has significantly lower (blue) or higher (red) intra-class similarity compared to ImageNet. As the figure suggests, using CLIP and ViT-based image embeddings yields highly consistent results. Furthermore, using other embeddings only strengthens our argument about the additional diversity in LAIONet.

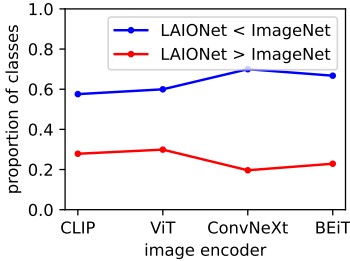

Figure 24: Comparing the intra-class similarity of LAIONet and ImageNet for various encoders. The vertical axis shows the proportion of classes in which one dataset has significantly lower intra-class similarity than the other, where we define significance at 95% confidence levels.

We also repeat our experiment in Section 4.2 using various image encoders. Remember that in Section 4.2, we generated new datasets A, B, C, and D by utilizing both the multimodal similarity between LAION images and synset text, as well as the textual similarity between LAION text and synset text. We then demonstrated in Figure 10c the fraction of classes where these datasets show lower intra-class similarity than ImageNet. In Figure 25, we show have repeated this experiment using other image embeddings. One can see that using any image encoder, going from dataset A to D, by increasing the role of multimodal similarity in dataset creation, the datasets become less and less diverse compared to ImageNet. This once again reaffirms our conclusion regarding the potential bias that the image-to-selection link can introduce in dataset creation.

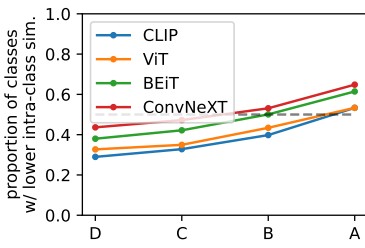

Figure 25: The proportion of classes where the new datasets A, B, C, and D in Section 4.2 have lower-intra class similarities than ImageNet, where we use various image encoders to calculate intra-class similarity.

# I    Sample Images From LAIONet

We provide randomly picked images from both CLIP-filtered and MPNet-filtered LAIONet (Appendix B) in this section. These images have been chosen based on various levels of difficulty. Figure 26 illustrates the distribution of the recall@5 difference of the ViT-base model for each common class between LAIONet and ImageNet. We choose recall@5 as a more reliable metric where the multiplicity of labels is less of a concern. One can see that there exist classes for which the recall on LAIONet is less than ImageNet for 0.5 or more. These are typically the classes for which LAIONet may have used a broader meaning for the synset or the images have appeared in a different context than ImageNet. It is worth noting that these classes make up a very small portion of all classes and have minimal impact on evaluations, whether or not including such images is desired.

For the classes labeled on the graphs of Figure 26, we have provided 10 random images from all datasets in the following. Each figure comes with a potential explanation for the failure of ImageNet models in the caption.

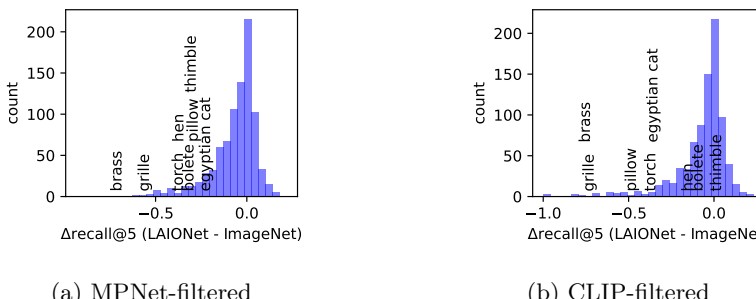

(a) MPNet-filtered                    (b) CLIP-filtered

Figure 26: Distribution of recall@5 on LAIONet subtracted by recall@5 on ImageNet. Only common classes are considered. The texts show the chosen classes for which example images are provided. The position of each text on the horizontal axis is the difference in recalls for that class.

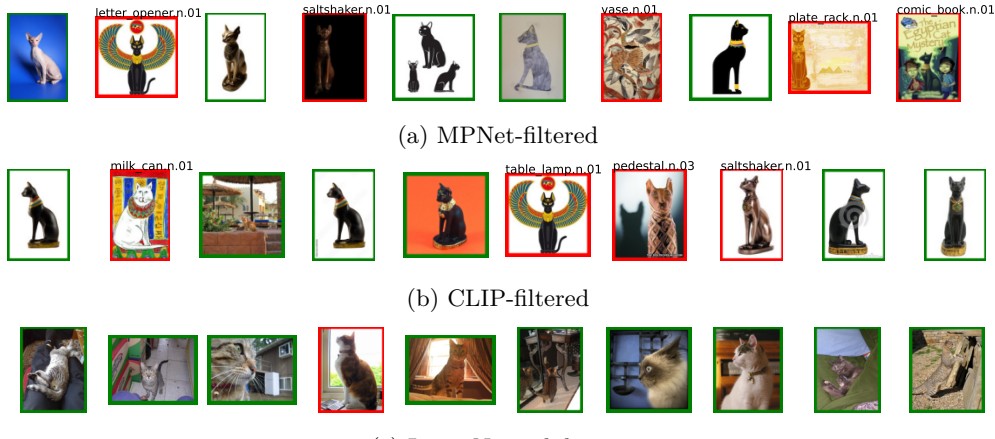

Figure 27: Egyption cat. ImageNet models primarily struggle with Egyptian cat statues or painted graphics, which are not well-represented or are rare in the ImageNet dataset.

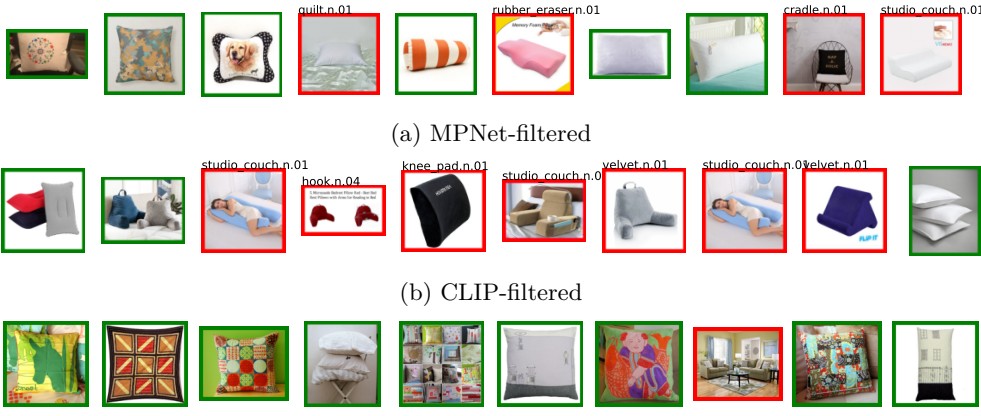

(a) MPNet-filtered

(b) CLIP-filtered

(c) ImageNet validation

Figure 28: Pillow. ImageNet models struggle to identify pillows when they deviate from the predominantly rectangular shape that is common in ImageNet.

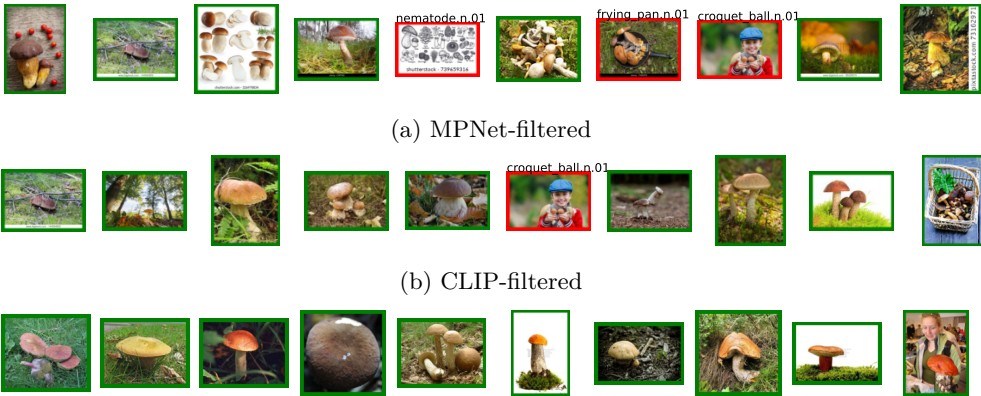

(a) MPNet-filtered

(b) CLIP-filtered

(c) ImageNet validation

Figure 29: Bolete. ImageNet models are challenged when a bolete appears in contexts outside of nature, such as being picked by a girl or found in a pan.

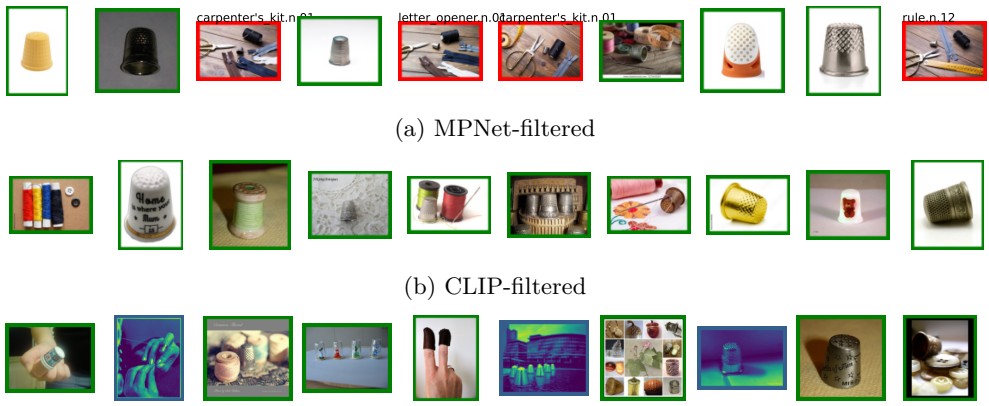

(a) MPNet-filtered

(b) CLIP-filtered

(c) ImageNet validation

Figure 30: Thimble. ImageNet models are challenged when the thimble is among many other items.

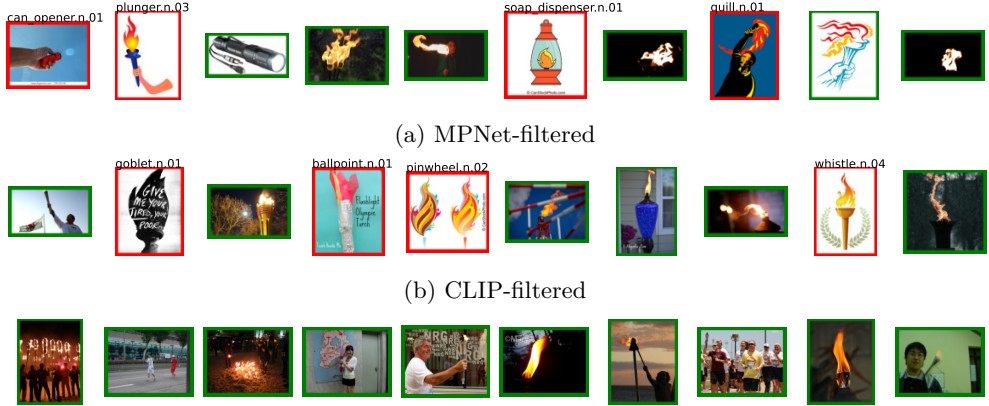

Figure 31: Torch. ImageNet models have difficulty with recognizing graphical depictions of torches and identifying variations in torch orientation.

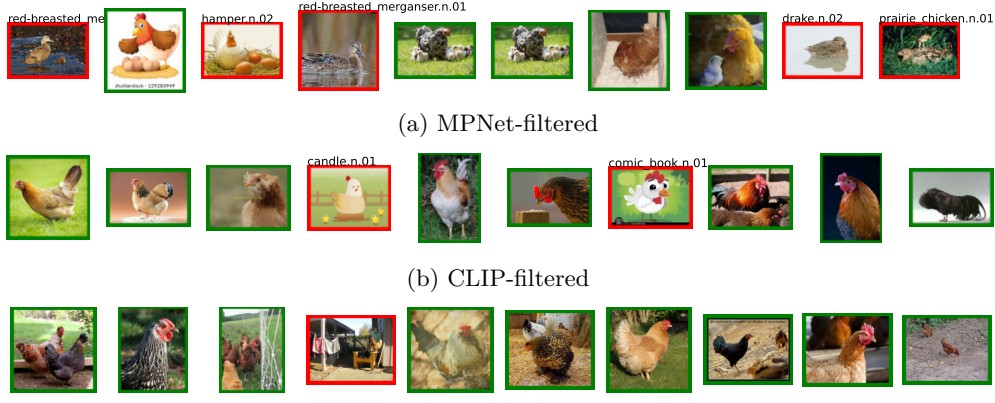

Figure 32: Hen. Graphical hens pose a challenge for ImageNet models. MPNet-filtered images also include blue and green-winged teal hens, which are not present in the ImageNet dataset.

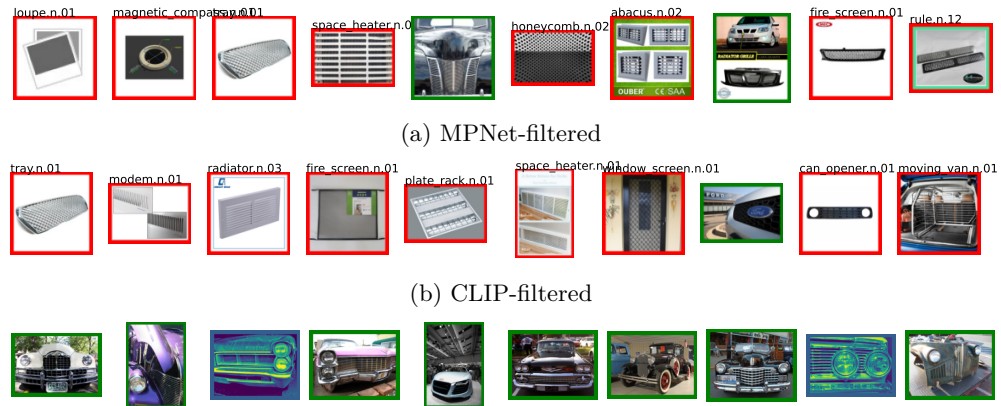

Figure 33: Grille. ImageNet models only recognize grille when installed on a car. LAIONet images also include various kinds of grille which are not meant by ImageNet class.

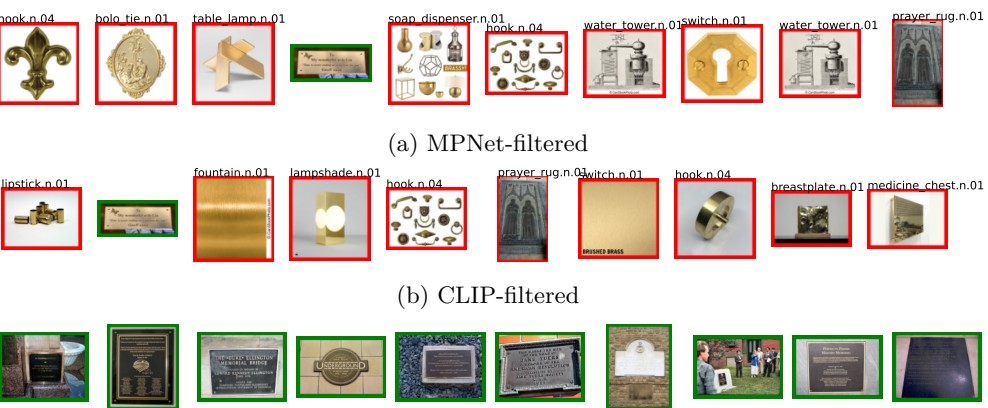

Figure 34: Brass. The intended concept of this class in ImageNet is a memorial made of brass. However, LAIONet images correspond to the broader meaning and the model is not expected to predict that.

