# OpenReview forum: "What Makes ImageNet Look Unlike LAION"
_TMLR — Accepted by TMLR_

### Review · Reviewer_Zheg · 2024-11-24

**Summary Of Contributions:**

This paper introduces LAIONet, a new dataset constructed by selecting images from the LAION dataset according  to ImageNet categories. Experimental results show the differences of model accuracy between LAIONet and ImageNet. This paper also an.alyzes the role of image-selection strategies in the dataset generation.

**Audience:**

Yes

**Claims And Evidence:**

Yes

**Requested Changes:**

Please refer to the weaknesses above.

**Strengths And Weaknesses:**

**Strengths:**
1. This paper investigates the differences between the widely-used LAION and ImageNet, which is interesting.
2. Analyzing image-to-selection link is a novel perspective to understand dataset differences.
3. The experiment and analysis may inspire future studies in related areas.

**Weaknesses:**
1. ImageNet V2 also leads to significant performance drop than ImageNet.  How is the trend in Figure 1 when comparing ImageNet V2 vs LAIONet?
2. LAIONet is selected according to text similarities between LAION text and a ImageNet synset using text encoders. Do CLIP and MPNet show biases in long-tail categories that has negative influence in data selection?
3. How many objects are typically in each image? Are there cases where an image includes multiple ImageNet categories?
4. Section 2 claims LAIONet images are not harder to classify from a CLIP perspective. Since LAIONet was curated using CLIP embeddings, does CLIP select easy images for itself to LAIONet?
5. Section 3.2 uses CLIP to evaluate visual similarity, but CLIP's vision encoder has known limitations [1]. How about using other vision models?
6. More details of how Figure 7(b)(c) is ploted are suggested.

[1] Tong et al. *Eyes Wide Shut? Exploring the Visual Shortcomings of Multimodal LLMs.* CVPR 2024.

---

> ### Author Response · Authors · 2024-12-19
>
> We thank the reviewer for their insightful and constructive feedback. We have updated the paper according to the reviewer’s suggestions and have addressed the comments and questions below.
>
> > How is the trend in Figure 1 when comparing ImageNet V2 vs LAIONet?
>
> **We have now included this analysis in Section 4.1.** Consistent with our earlier findings, ImageNet-trained models demonstrate comparable accuracy on LAIONet and ImageNetV2-b, the version of ImageNetV2 closest to LAIONet in terms of intra-class similarity. As the diversity of images decreases in ImageNetV2-a and ImageNetV2-c, the models' performance improves. We thank the reviewer for this suggestion and believe this additional analysis has enriched Section 4.1.
>
> > Do CLIP and MPNet show biases in long-tail categories that has negative influence in data selection?
>
> We acknowledge that a biased text encoder could influence our selection adversely. However, we are not currently aware of any well-documented biases in the text encoders we used, in particular, the MPNet-based encoder. In general, we expect text encoders to be less biased than vision encoders, as the visual domain is a much richer and more complex space, and we believe this is why LAIONet contributes more diverse images.
>
> > How many objects are typically in each image? Are there cases where an image includes multiple ImageNet categories?
>
> **We have included additional statistics of the dataset in our revised manuscript.** Regarding the number of images per class, 95% of LAIONet classes have at least 6 samples, 90% have at least 13 samples, 80% have at least 36 samples, and 70% have at least 75 samples. To simplify evaluations, we have excluded images assigned to multiple classes in LAIONet.
>
> > Section 2 claims LAIONet images are not harder to classify from a CLIP perspective. Since LAIONet was curated using CLIP embeddings, does CLIP select easy images for itself to LAIONet?
>
> We agree with the reviewer that, since the CLIP text encoder used in the creation of LAIONet is jointly trained with its vision encoder, which is used in our argument about the difficulty of LAIONet, this could potentially give CLIP an advantage on LAIONet. We have acknowledged this limitation in the last paragraph of Page 5. However, it is important to note that this argument is not central to our main analysis of ImageNet selection bias; we mainly discuss this notion of difficulty for the completeness of our analysis. **We have rephrased the last paragraph of Page 5** to more clearly communicate this limitation and thank the reviewer for highlighting it.
>
> > Section 3.2 uses CLIP to evaluate visual similarity, but CLIP's vision encoder has known limitations. How about using other vision models?
>
> We thank the reviewer for their suggestion and for highlighting the shortcomings of CLIP’s vision encoder. To ensure that our analysis of intra-class similarities is not confined to CLIP, **we have repeated these calculations using a variety of encoders, as detailed in Appendix H**. The new results confirm that our findings regarding LAIONet’s additional diversity remain consistent across all visual encoders that we tried. We appreciate the reviewer’s feedback and believe these observations further strengthen the basis for our claims.
>
> > More details of how Figure 7(b)(c) is plotted are suggested.
>
> We thank the reviewer for this suggestion. In the updated manuscript, we have further elaborated on these plots in the caption of Figure 7.

---

> > ### Comment · Reviewer_Zheg · 2025-01-09
> >
> > Thanks to the authors for the additional analysis and experiments, which have addressed my concern.

---

### Review · Reviewer_DWnV · 2024-11-24

**Summary Of Contributions:**

The submission presents a new dataset LAIONet that is sourced from LAION 400M by searching for lemmas from the imagenet net classes' synsets in the LAION text attributes, and then further filtering these candidates with a fixed threshold by their CLIP text similarity to the synset text.

Models trained on ImageNet-1k/22k perform worse on LAIONet, and LAIONet has a lower intra-class similarity than ImageNet.

The authors propose that this difference can be explained by the data generation process, namely that the selection criteria of LIONet is limited to the text vs a selection criteria that involves both text and image in the case of ImageNet.

**Audience:**

Yes

**Claims And Evidence:**

No

**Requested Changes:**

1. `[critical]` I would want to see a **controlled setup**, where only specific parts of the dataset generation process are changed, and then the resulting datasets are compared against each other. If the authors want to show that higher intra-class similarity is due to a selection condition based on both the image and the text, vs. only the text, then they should define an additional dataset generation process that shares all characteristics with the dataset generation process of LAIONet, but changes the selection criteria, and then compare the resulting datasets.

1. Provide **additional LAIONet characterization**, such as the number of examples per class.

1. The authors mention multiple times that LAIONet contains "images that are easy to classify" (Section 1.1), and that "at least from the CLIP view, LAIONet images are not harder to classify." (Section 2). But images could actually be quite hard to classify, for example because the text does not describe the image well in some cases. It would be interesting to have a **more detailed analysis of the mis-classifications** from different models. I found some examples in Figures 21-29, but these classes are not representative for the majority of the images in LAIONet.

1. The **filtering** is crucial for the generation of LAIONet. The paper should explain why examples were first filtered by presence of a lemma in the text, and then by applying a threshold to the CLIP text-similarity of the example text and name and definition of the synset. Why is the first step necessary (the text only argues why the second step is necessary)? What happens to the intra-class similarity when the threshold is changed?

Additional Questions

1. The first phrase of the abstract states that "ImageNet was famously created from Flickr image search results." Where exactly is this described? Looking at [(Deng et al, 2009)] I read "We collect candidate images from the Internet by querying several image search engines" in Section 3.1
1. Why was LAION 400M used and not LAION 5B? Having a larger dataset would allow for better class coverage and more freedom in the selection procedure.
1. Figure 2: What is the meaning of some lines being dashed?
1. Figure 3b: "ImageNet classes covered" – covered with how many examples?

[(Deng et al, 2009)]: https://www.image-net.org/static_files/papers/imagenet_cvpr09.pdf

**Strengths And Weaknesses:**

Strengths

1. The paper presents a new dataset that can be used to evaluate ImageNet models, with the same classes but significantly more varied images.

1. The authors make use of pre-existing ImageNet datasets in their analysis, like ImageNet-Captions, and ImageNetV2.

1. The authors come up with a hypothesis about the data generating process that could account for the observed difference, and the submission describes multiple experiments that try to give support to that hypothesis.

Weaknesses

1. The paper compares two datasets that were generated in completely different ways: ImageNet was using various image search engines to find images matching the textual descriptions, LAIONet sources images exclusively from a dataset that is already prefiltered in various ways, one of them being a CLIP image/text similarity threshold; ImageNet being collected from the web before the year 2009, LAIONet containing images up to 2021; ImageNet being selected by human rater agreement, LAIONet completely relying on a CLIP-based selection process to remove noise; (and the list goes on...) But the author's central claim is that the observed difference of the two datasets (namely its lower intra-class similarity) is due to information bottleneck in the selection criteria, and the higher intra-class variance of LAIONet being due to the fact that only the text was used for the selection process. I agree that this is a possible source of the observed difference, but the paper is lacking a quantification of **how much of the observed difference is due to this difference vs. all the other differences** in the dataset generation process.

1. The introduced dataset LAIONet lacks some detail. For example, I only found Figure 4 about the class frequencies, and that figure only has approximate information about the 500 most frequent classes (for the least frequent classes, how often do they actually appear in the dataset?).

---

> ### Author Response · Authors · 2024-12-19
> **Part 1 / 2**
>
> We thank the reviewer for their insightful comments and helpful feedback. In response, we have updated the paper based on the reviewer’s suggestions and have addressed the questions and requested changes in the following.
>
> > If the authors want to show that higher intra-class similarity is due to a selection condition based on both the image and the text, vs. only the text, then they should define an additional dataset generation process that shares all characteristics with the dataset generation process of LAIONet, but changes the selection criteria, and then compare the resulting datasets.
>
> We thank the reviewer for this suggestion. We agree that a controlled setup can make our arguments stronger. **Therefore, we have included an additional analysis following the reviewer’s suggestion in the new Section 4.2**. Here, we create new versions of LAIONet using both the image and text in the selection process. In particular, we require image-to-[name: def] similarity to be sufficiently large in addition to text-to-[name: def] similarity. We choose the similarity thresholds while controlling for the total number of selected samples. Looking at the intra-class similarity of these new datasets, we can clearly see that using visual information during selection can reduce the diversity of the selected images. To ensure our measurement of intra-class similarity is not specific to CLIP, we have also repeated this analysis in Appendix H using a variety of image encoders and found consistent results.
>
> We should emphasize that the new experiment mostly shows how relying on a richer modality can make the selection process more susceptible to biases. Specifically, it highlights how using visual information through multimodal similarity calculation can introduce bias during selection. However, this experiment does not directly address how visual information was used in the creation of ImageNet. That question is explored in Section 4.1, where we controlled the strictness of human labeling criteria and observed that the diversity of ImageNet-V2 changes with the strictness level. Together, we believe these experiments provide two controlled setups explaining how an image-to-selection link can be introduced during dataset creation and the significant impact such a link can have on the diversity of the included images. We thank the reviewer for this suggestion and believe the new experiment has enriched our paper.
>
> > Provide additional LAIONet characterization, such as the number of examples per class.
>
> **We have included additional statistics of the dataset in our revised manuscript**: 95% of LAIONet classes have at least 6 samples, 90% have at least 13 samples, 80% have at least 36 samples, and 70% have at least 75 samples. Also, note that none of the included images has been assigned with two labels. When it comes to the calculation of intra-class similarity, we have only considered the classes that have at least 7 images so we can calculate at least 49 pairwise similarities and do a reliable comparison. We have explained these nuances in the revised text and thank the reviewer for the suggestion.
>
> > ... images could actually be quite hard to classify, for example because the text does not describe the image well in some cases. It would be interesting to have a more detailed analysis of the mis-classifications from different models.
>
> We agree that LAIONet includes images that are challenging to classify, such as those where the object is not central to the image (Fig. 30), appears in new formats (Fig. 28), or is situated in unfamiliar contexts (Fig. 33). When we stated that LAIONet images are not harder to classify for CLIP, we mostly referred to the observation that the average accuracy on ImageNet and LAIONet is comparable. However, we do not discount the presence of difficult-to-classify images within LAIONet. We thank the reviewer for raising this clarifying question and revised the text accordingly.
>
> Regarding an analysis of misclassifications, we have included various examples from classes where a base ViT model has the largest performance drops on LAIONet in Appendix I. That said, we agree that a deeper investigation into the misclassifications of different models could be highly valuable. Our aim is to highlight the root cause of these misclassifications---namely, the selection bias we have identified---and bring this issue to the center of community discussions. Given that LAIONet will be publicly available for download and reproduction, we hope this will encourage further exploration of these issues.
>
> ...

---

> ### Author Response · Authors · 2024-12-19
> **Part 2 / 2**
>
> > ... why examples were first filtered by presence of a lemma in the text, and then by applying a threshold to the CLIP text-similarity of the example text and name and definition of the synset.
>
> Requiring the presence of lemmas in the text helps ensure that we are not misclassifying ImageNet classes that are semantically very similar. Analyzing the CLIP textual similarity of distinct pairs of synsets, we find that 5% of all pairs have a similarity of 0.72 or higher. While our threshold of 0.82 is generally effective in distinguishing most of these pairs, it cannot fully separate them. For example, consider the extreme case of “miniature poodle” and “toy poodle,” two ImageNet classes whose [name: def] has a high similarity of 0.9. In such cases, it is necessary to ensure that the lemma is explicitly mentioned in the text.
>
> **Additional Questions**
>
> > The first phrase of the abstract states that "ImageNet was famously created from Flickr image search results." Where exactly is this described?
>
> We thank the reviewer for bringing this to our attention. We fixed this in the abstract. ImageNet-V2 paper mentioned their dataset collection follows “the original process of utilizing the Flickr image hosting service,” however we were not able to find any other official comments on ImageNet, so we decided to use the general term of the “image search engines” in our updated paper.
>
> > Why was LAION 400M used and not LAION 5B? Having a larger dataset would allow for better class coverage and more freedom in the selection procedure.
>
> We agree that LAION 5B could offer even better coverage of classes. However, we largely worked with LAION 400M for convenience, as it allowed us to generate different versions of LAIONet quickly, as discussed in response to the reviewer, and to run as many experiments as possible.
>
> > Figure 2: What is the meaning of some lines being dashed?
>
> The dashed lines represent the possibility that these links exist. In particular, whether an effect from X to T (or vice versa) exists does not change our conclusions. We now have clarified this in the caption and thank the reviewer for the question.
>
> > Figure 3b: "ImageNet classes covered" – covered with how many examples?
>
> In this figure we just mean that a class has at least one example. Note that more than 95% of the classes have at least six samples, though. We have not clarified this in the caption.

---

> > ### Comment · Reviewer_DWnV · 2025-01-07
> >
> > Thank you for the detailed replies to my questions.
> >
> > I find especially the added section 4.2 interesting, and I agree that it partly addresses my main concern about the controlled setup. While this added section still cannot conclusively show that the observed differences with ImageNet are indeed due to the image-selection criteria (and not some other difference between the datasets, see my initial comment), it certainly does give more evidence pointing towards this hypothesis, and I agree that together with the data presented in Section 4.1 this gives enough support for the central claim of the paper.

---

### Review · Reviewer_2Yxa · 2024-12-10

**Summary Of Contributions:**

The paper introduces LAIONet, a dataset from LAION-400M based on textual similarity to ImageNet, supporting a counterfactual analysis of ImageNet's structure. It reveals differences between LAIONet and ImageNet, such as LAIONet's lower intra-class similarity, which causes ImageNet-trained models to perform worse on it. They further provide causal explanations for these differences, attributing ImageNet's higher intra-class similarity to human annotation, which introduces selection bias, whereas LAIONet reflects more natural data distributions due to its text-based selection. This study offers actionable insights for dataset creation, advocating for the use of information bottlenecks like text captions to reduce bias and improve diversity in datasets.

**Audience:**

Yes

**Broader Impact Concerns:**

It might be helpful to elaborate further on the process used to filter out NSFW content from LAION-400M and discuss the measures taken to make sure those content is excluded from LAIONet.

**Claims And Evidence:**

Yes

**Requested Changes:**

- Potentially including some small-scale ablation studies across different similarity thresholds and providing empirical justification for the chosen values would be helpful to understand why 0.3 is a good threshold
- To properly analyze the performance gap, I suggest that the authors conduct bidirectional experiments by training models on LAIONet and testing them on ImageNet.

**Strengths And Weaknesses:**

Strength:
1. This paper is well-written, well-organized, and well-motivated, tackling a fundamental issue in vision dataset creation.
2. The authors provide comprehensive experiments comparing LAIONet and ImageNet, evaluating accuracy, intra-class similarity, and the causal data-generation processes.
3. The study aims to understand the role of human annotation through ImageNetV2 analysis, and the relationship between text captions and image selection. They compare ImageNetV2 and ImageNet-Captions to support their claims about dataset biases and diversity.
4. Additionally, they offer insightful analysis of the selection approaches on dataset structure. These strengths make the work a valuable contribution to understanding and improving dataset curation.

Weakness:
1. Using textual similarity as the sole selection criterion may oversimplify the multimodal factors in dataset creation and fail to account for contextual biases in text captions.
2. While the paper shows LAIONet has greater diversity, it doesn't show whether this diversity always represents meaningful variations or might include some noisy/irrelevant variations.
3. Although the authors propose using text as an information bottleneck for future dataset creation, they don't provide concrete guidelines about optimal text similarity thresholds or how to balance coverage versus precision in text-based filtering. Some hyperparameter settings seem very arbitrary, for example “excluding instances with a cosine similarity of image and text embeddings less than 0.3.” Why 0.3? Why not higher/lower? Also “we require a high similarity between the text and the synset name and definition” but what if the original LAION caption doesn’t include the relevant information at all? How to handle this situation?
4. The reported 5-12 percentage point accuracy drop when evaluating ImageNet models on LAIONet could be attributed to dataset shift rather than intra-class similarity differences. Is there any experiment on the other way around? For example testing out the models trained with LAIONet on ImageNet.

---

> ### Author Response · Authors · 2024-12-19
> **Part 1 / 2**
>
> We thank the reviewer for their detailed feedback and insightful questions. We are glad that the reviewer found our study well-written, well-motivated, and a valuable contribution to understanding dataset curation! In the following, we address the comments and questions raised by the reviewer.
>
> > Using textual similarity as the sole selection criterion may oversimplify the multimodal factors in dataset creation and fail to account for contextual biases in text captions.
>
> We agree that text-based selection can also be affected by contextual biases, as the reviewer has suggested. However, our main point is that text and text-based selection are generally less susceptible to bias than richer modalities, such as images. Our various experiments support this intuition: when we use an information bottleneck like text in our selection process, we are able to create a more diverse dataset.
>
> > While the paper shows LAIONet has greater diversity, it doesn't show whether this diversity always represents meaningful variations or might include some noisy/irrelevant variations.
>
> While we cannot dismiss the existence of noisy labels, we believe that our selection criteria work to minimize their presence. In particular, in Appendix I of our revised manuscript, we have included random images from the classes where the ViT-base model performs significantly worse on LAIONet. These classes are more likely to contain significant noise. Looking at Fig. 26, the difference in recall between LAIONet and ImageNet shows a long tail. In the majority of cases, this difference is less than 0.5. Our examples illustrate that such a drop can be attributed to the diversity of LAIONet images and labels are accurately assigned. For example, in Fig. 27, the model struggles with Egyptian cat statues or painted graphics, which are rare in ImageNet. Or in Fig. 28, the model struggles to recognize pillows when they deviate from the rectangular shape that is common in ImageNet. Figures 29-34 provide additional examples. In the rare instances where the difference in recalls exceeds 0.5, it may be because LAIONet has used a broader meaning for the class (such as “brass” in Fig. 34), or the images have appeared in a different context than in ImageNet. It’s a good question whether including these images in LAIONet is desired or not, but in any case, these classes constitute a very small portion of all classes and have minimal impact on evaluations.
>
> > ... provide concrete guidelines about optimal text similarity thresholds or how to balance coverage versus precision in text-based filtering.
>
> Regarding guidelines for the optimal textual similarity threshold, the higher the threshold, the more confident we can be in the quality of the match. Therefore, in Sec. 2, particularly in Fig. 3, we chose the highest threshold that still covers a significant portion of ImageNet classes. While we opted for the largest possible threshold to be on the safe side, we also offer some insights into what threshold might be sufficient. For example, in Sec. 4 (Fig. 8b), using the restored captions of ImageNet, we show that a textual similarity above 0.7 is often enough to ensure that a sample is uniquely associated with the synset.
>
> > ... what if the original LAION caption doesn’t include the relevant information at all? How to handle this situation?
>
> If the caption lacks relevant information, we simply exclude that sample. While this could potentially lead to a false negative, for dataset generation from a massive dataset like LAION, our primary goal is to use a method with a very small number of false positives. In other words, we only need to ensure that the label is correct, and it’s ok to exclude a sample because we can’t label is with certainty.
>
> > The reported 5-12 percentage point accuracy drop when evaluating ImageNet models on LAIONet could be attributed to dataset shift rather than intra-class similarity differences. Is there any experiment on the other way around? For example testing out the models trained with LAIONet on ImageNet.
>
> One such evidence is Fig. 23 in Appendix G.2 of the revised paper, where we show that ImageNet-trained models perform well on the classes where LAIONet has similar or higher intra-class similarity than ImageNet. Another piece of evidence comes from Fig. 8, where the models perform worse on the ImageNet-V2 versions that are more diverse due to a lower MTurk selection frequency. As we discuss in response to the requested changes in the following, we currently do not have any evidence from LAIONet-trained models.
>
> ...

---

> ### Author Response · Authors · 2024-12-19
> **Part 2 / 2**
>
> > Potentially including some small-scale ablation studies across different similarity thresholds and providing empirical justification for the chosen values would be helpful to understand why 0.3 is a good threshold
>
> We thank the reviewer for this suggestion. To clarify, the 0.3 threshold refers to the image-to-caption CLIP similarity used by the LAION authors to include only relevant image-caption pairs. Our choice of threshold is the minimum textual similarity required to include a sample in LAIONet. We selected this threshold to be the largest possible one while still maintaining coverage across classes. Furthermore, we demonstrate that this 0.82 threshold is significantly higher than the 0.7 threshold suggested by our analysis in Sec. 4 (Fig. 8b). We provide an example of how the output varies with different thresholds in Appendix C of the updated manuscript.
>
> > To properly analyze the performance gap, I suggest that the authors conduct bidirectional experiments by training models on LAIONet and testing them on ImageNet.
>
> We thank the reviewer for this suggestion. Bidirectional experiments would certainly be interesting. Unfortunately, due to the limited time available for rebuttal, we were unable to conduct these additional experiments. In general, since we created LAIONet to contrast ImageNet and LAION, we believe such experiments may not substantially contribute to our conclusions, which already indicate that LAIONet images are more diverse. As a result, we chose to focus our efforts in the paper on understanding the underlying cause of this difference.
>
> > It might be helpful to elaborate further on the process used to filter out NSFW content from LAION-400M and discuss the measures taken to make sure those content is excluded from LAIONet.
>
> We used the NSFW tags provided by LAION curators and filtered out "unsafe" and "unlikely" samples.

---

### Author Response · Authors · 2024-12-19

We thank the reviewers for their engaged, insightful, and constructive reviews. Below, we address each of the questions raised by the reviewers and have uploaded a new version of the work, in line with the reviewers’ feedback. We have highlighted changes to the text in blue.

---

### Decision · Action_Editor_Yk5Z · 2025-01-14

**Recommendation:** Accept as is

**Comment:**

The paper is of clear interest to the community. All three reviewers are convinced by the revised version.

**Audience:**

Yes. The topic of how datasets come to be is of clear interest to the TMLR community. Both datasets involved in the paper (LAION and Imagenet) are foundational datasets for several fields.

**Claims And Evidence:**

Yes. Post-discussion all reviewers agree that the paper supports its claims with accurate, convincing, and clear evidence.